# Heterogeneous receptor expression underlies non-uniform peptidergic modulation of olfaction in *Drosophila*

Tyler R. Sizemore [1,3] ✉, Julius Jonaitis [1] & Andrew M. Dacks [1,2] ✉

Sensory systems are dynamically adjusted according to the animal's ongoing needs by neuromodulators, such as neuropeptides. Neuropeptides are often widely-distributed throughout sensory networks, but it is unclear whether such neuropeptides uniformly modulate network activity. Here, we leverage the *Drosophila* antennal lobe (AL) to resolve whether myoinhibitory peptide (MIP) uniformly modulates AL processing. Despite being uniformly distributed across the AL, MIP decreases olfactory input to some glomeruli, while increasing olfactory input to other glomeruli. We reveal that a heterogeneous ensemble of local interneurons (LNs) are the sole source of AL MIP, and show that differential expression of the inhibitory MIP receptor across glomeruli allows MIP to act on distinct intraglomerular substrates. Our findings demonstrate how even a seemingly simple case of modulation can have complex consequences on network processing by acting non-uniformly within different components of the overall network.

Animals use their sensory systems to internalize and process information about the identity, intensity, and valence of external stimuli, so they can properly navigate their environment. However, constant ecological and internal state fluctuations threaten the animal's ability to accurately represent these stimuli. To address the demands these fluctuations impose, sensory systems use processes such as neuromodulation to flexibly adjust sensory processing and behavior. The largest and most ancient collection of neuromodulators are small peptides (~3–100 amino acids) termed neuropeptides[1–10]. For instance, neuropeptide F (NPF)/neuropeptide Y (NPY) play a conserved role in promoting feeding behaviors in sea slugs, humans, flies, zebrafish, nematodes, mosquitoes, and rodents[11–19]. Yet, despite their ubiquity and clear importance in nervous systems[20–22], the mechanistic basis of peptidergic modulation of sensory processing remains unclear.

Often peptidergic modulation is strongly associated with a given physiological drive, and in some cases the actions of a neuropeptide within a single network can be associated with different behavioral contexts. For instance, myoinhibitory peptide (MIP) is necessary and sufficient to stimulate the drive of *Drosophila* towards food-odors

within the context of satiation[23], and is implicated in the post-mating shift in odor-preferences[24]. However, MIP appears to be uniformly distributed across all antennal lobe (AL) glomeruli, which process far more than just food-related odors. Therefore, how can a ubiquitously distributed neuropeptide have stimulus-specific effects on sensory processing? Here, we reveal the cellular, physiological, and structural substrates that enable MIP to differentially modulate olfactory input to distinct olfactory channels.

## Results

### MIP differentially modulates olfactory input to distinct glomeruli

To test whether odorant responses within different glomeruli are uniformly modulated by MIP, we chose an odor which activates several glomeruli visible at the same/nearly the same imaging depth, and whose pattern of glomerular activation is well-established—apple cider vinegar (ACV)[25]. In this way, any non-uniform effects of MIP on OSN odor-responses across different glomeruli would be readily detectable. Furthermore, we recorded the odor-evoked responses of OSN

[1]Department of Biology, Life Sciences Building, West Virginia University, Morgantown, WV 26506, USA. [2]Department of Neuroscience, West Virginia University, Morgantown, WV 26506, USA. [3]Present address: Department of Molecular, Cellular, and Developmental Biology, Yale Science Building, Yale University, New Haven, CT 06520-8103, USA. ✉e-mail: tyler.sizemore@yale.edu; Andrew.Dacks@mail.wvu.edu

axon terminals (i) before MIP was pressure injected onto the AL ("pre-MIP injection"), (ii) after MIP was injected, but before it was removed from the perfusate ("MIP"), and (iii) after a brief washout period ("post-washout") (see Methods) (Fig. 1a).

Before MIP application, OSNs robustly respond to both test concentrations of ACV (Fig. 1b, c), then after MIP is pressure injected into the AL DM1 OSN responses to $10^{-2}$ ACV are increased (Fig. 1b). Similarly, DM4 OSN responses to $10^{-6}$ ACV are also increased after peptide application (Fig. 1c). After a brief washout period, DM1 OSN responses to $10^{-2}$ ACV return to pre-peptide application responses (Fig. 1b), whereas the increased DM4 OSN responses to $10^{-6}$ ACV are sustained (Fig. 1c). In contrast, DM2 OSN responses are substantially diminished after peptide application regardless of odor concentration, and remain so post-washout (Fig. 1b, c). Moreover, DM5 OSN responses to $10^{-2}$ ACV, which were decreased (albeit insignificantly) upon peptide application, become significantly diminished post-washout relative to pre-peptide application (Fig. 1b).

Altogether, these results show that MIP can differentially modulate OSN odor-evoked responses in a glomerulus- and stimulus

concentration-dependent manner, while also having concentration-independent consequences on OSNs of another glomerulus. However, these observations could be explained by differences in MIP-SPR signaling substrates across these glomeruli. For instance, there may be no synaptic input to DM1 OSNs from MIP-immunoreactive (MIP-ir) AL neurons, and therefore our prior observations (Fig. 1b, c) are the result of polysynaptic influences induced by MIP application. Therefore, we sought to test our suppositions by resolving the entire MIPergic signaling circuit architecture, including the identity of the presynaptic MIP-releasing AL neurons, their pre- & postsynaptic partners, and those postsynaptic partners that express SPR.

## Patchy GABAergic LNs are the sole source of MIP within the *Drosophila* AL

Previous neuroanatomical investigations suggested that the neurites of the AL-associated MIPergic neurons appear restricted to the AL, which implies MIP is released from AL LNs[26]. However, the *Drosophila* AL houses ~200 LNs whose distinct roles in olfactory processing have been associated with their transmitter content and morphology[27–37].

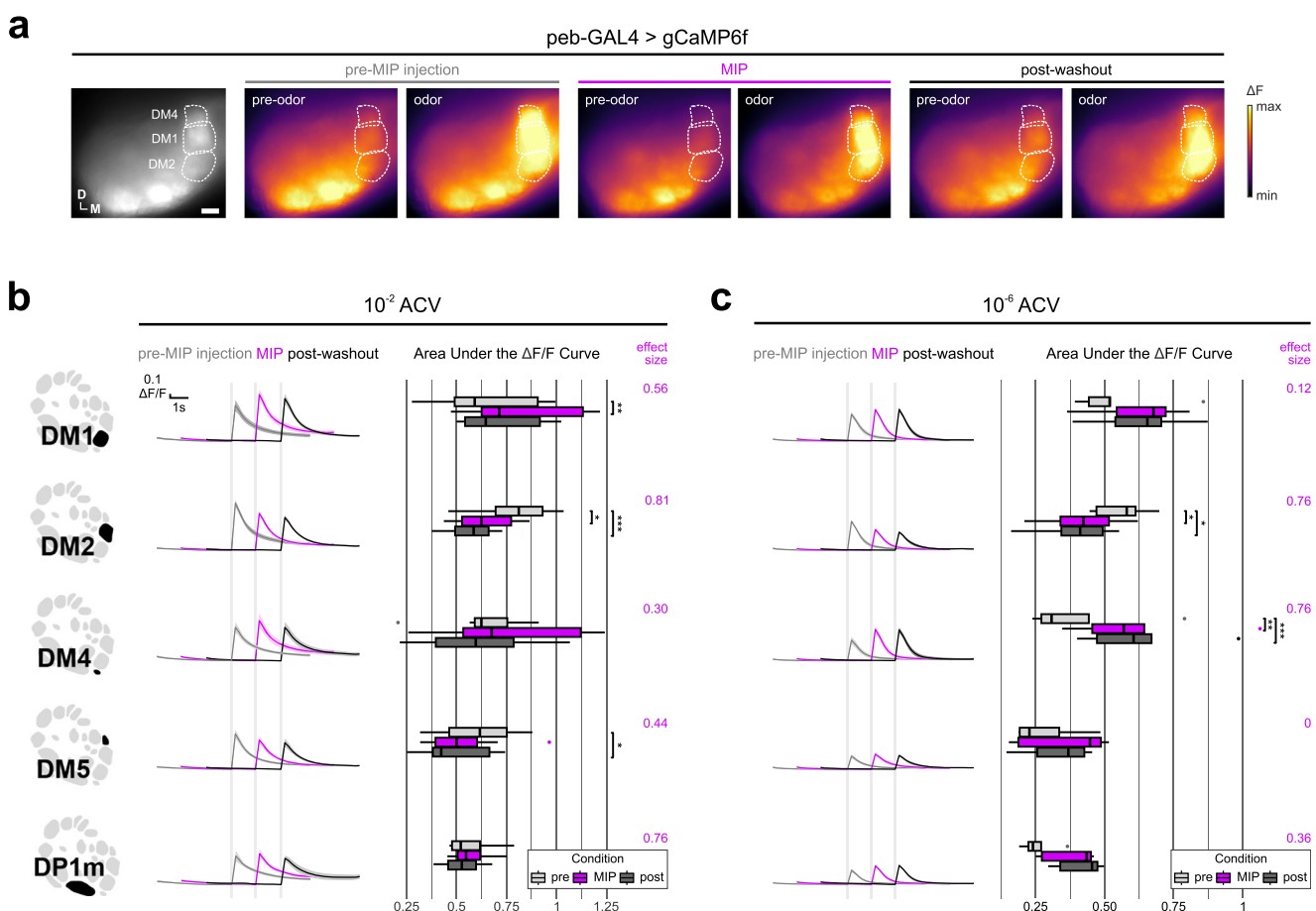

**Fig. 1 | Myoinhibitory peptide (MIP) differentially modulates OSN ACV responses. a** Representative pseudocolored heatmaps of OSN GCaMP responses (ΔF) before and during odor presentation in several test glomeruli (white dotted outlines). In each case, each odor presentation heatmap pair is grouped by stage of MIP pharmacological application. Scale bar = 10 μm. **b** DM1, DM2, DM4, DM5, and DP1m OSN responses to $10^{-2}$ ACV before (pre-MIP injection), after MIP pressure injection (MIP), and post-washout (post-washout). MIP increases DM1 OSN responses ($p = 0.004$, pre-MIP vs. MIP; $n = 8$ animals). Conversely, MIP decreases DM2 and DM5 OSN responses (DM2: $p = 0.013$, pre-MIP vs. MIP & $p = 0.001$, pre-MIP vs. post-washout; $n = 8$ animals; DM5: $p = 0.02$, pre-MIP vs. post-washout; $n = 8$ animals). **c** Same as **b**, but in response to $10^{-6}$ ACV. MIP decreases DM2 OSN responses ($p = 0.013$, pre-MIP vs. MIP & $p = 0.01$, pre-MIP vs. post-washout; $n = 5$

animals). Conversely, MIP increases DM4 OSN responses ($p = 0.002$, pre-MIP vs. MIP & $p = 0.001$, pre-MIP vs. post-washout; $n = 5$ animals). In all cases: Data are presented as the mean (darker center line) ± SEM (lighter shaded area). Vertical & horizontal scale bars = 0.1 ΔF/F & one second (respectively). Odor onset is indicated by vertical lines running up each column of traces. Boxplots display the minimum, 25th-percentile, median, 75th-percentile, and maximum of the given data. Statistical significance was assessed using a two-tailed pairwise repeated measures t-tests (RM t-tests) with a Holm multiple comparisons correction. Statistical measures of effect size (Cohen's *d*) are provided to the right of each set of AUC boxplots. All statistical tests were two-tailed. Glomerular schematics were derived from an in vivo AL atlas[164]. Source data are provided as a Source Data file.

For example, individual cholinergic AL LNs innervate many glomeruli and perform lateral excitation as a means for broadening odor representations in the AL[33,34,38–40]. Therefore, to resolve whether MIP-ir AL neurons are indeed AL LNs, and if they belong to a known AL LN chemical class, we assessed the overlap of MIP-immunoreactivity with markers for the major *Drosophila* small-neurotransmitters[41] (Fig. 2a and Supplementary Fig. 1). We find that AL MIP-ir neurons do not overlap with choline acetyltransferase (ChAT) or vesicular glutamate transporter (VGlut), but all MIP-ir neurons in the AL overlap with GAD1 (9.1 ± 0.19 neurons, $n = 5$ brains, 10 ALs) (Fig. 2a and Supplementary Fig. 1). In accordance with RNA-sequencing[42–44], we find no detectable MIP-immunoreactive OSNs (Supplementary Fig. 1). Altogether, these results suggest that an ensemble of ~9 GABAergic LNs are the source of MIP within the *Drosophila* AL.

The *Drosophila* AL houses a variety of distinct GABAergic LNs, which can be subdivided into five major morphological types: pan-glomerular, multiglomerular, oligoglomerular, continuous, and patchy[32]. Like cortical interneurons[45,46], these different interneuron morphological types play distinct roles in AL olfactory processing. To determine morphological type to which the MIPergic LNs belong, we screened the Janelia FlyLight driver line collection[47], tested ~25 of those lines for MIP-immunoreactivity, and identified a GAL4 driver (R32F10-GAL4) that selectively highlights MIPergic LNs within the AL (Fig. 2b–d). We then combined R32F10-GAL4 with a GFP-tagged rat preproatrial natriuretic factor (ANF-GFP) which, when expressed within peptidergic neurons, is proteolytically processed and packaged into secretory vesicles and preferentially accumulates in peptidergic synaptic terminals[48]. We find broad ANF-GFP accumulation in R32F10-GAL4 AL LN terminals across the AL, confirming these LNs possess the necessary subcellular machinery for neuropeptide processing, packaging, and release (Fig. 2e). Furthermore, all MIP-ir is abolished in the AL when R32F10-GAL4 AL LNs are ablated via temperature-gated expression of a cell-specific diphtheria toxin (Fig. 2f). We then leveraged our selective genetic access to these LNs to resolve the morphology of individual MIPergic LNs through stochastic labeling[49]. From these experiments, we find that all MIPergic LNs have a discontinuous innervation pattern resembling that of patchy AL LNs (Fig. 2g and Supplementary Fig. 1). Taken together, our data suggests that ~9 GABAergic patchy LNs are the sole source of MIP within the *Drosophila* AL.

There are many AL neurons, including other LNs, that are not patchy LNs but resemble the discontinuous morphology described for patchy LNs[32]. However, individual patchy LNs are unique in that they innervate different sets of glomeruli from animal-to-animal[32]. Therefore, we analyzed the set of glomeruli innervated by 50 individual MIPergic LNs and observed 50 distinct innervation patterns, thus demonstrating that no individual MIPergic LN innervates the same set of glomeruli across animals (Fig. 2h and Supplementary Figs. 2 and 3). Additionally, we find individual MIPergic LNs do not preferentially innervate any one glomerulus over others (Supplementary Fig. 2). When sister clones were assessed, we find that two individual MIPergic LNs co-innervate ~12 glomeruli ($n = 5$ brains, 5 sister clones) (Supplementary Fig. 3a, b), and individual MIPergic LNs consistently innervated at least one of the hygro-/thermosensory associated glomeruli (Supplementary Fig. 3c–e). These results suggest that at least two MIPergic LNs innervate any single glomerulus, including hygro-/thermosensory glomeruli[50–52]. Moreover, these observations also demonstrate that individual MIPergic LNs innervate different glomeruli from animal-to-animal.

Most odorants activate more than one glomerulus in the *Drosophila* AL[25,53–55]. Thus, if individual MIPergic LNs innervate different sets of glomeruli from animal-to-animal, are there pairs of glomeruli that are innervated significantly more than others? If so, what ecological relationships exist amongst significantly correlated pairs of glomeruli? To determine the probability that an individual MIPergic LN that

innervates one glomerulus will innervate/avoid another glomerulus, we leveraged our previous clonal analysis data (Fig. 2g, h) to calculate a correlation coefficient for all possible pairs of glomeruli (Fig. 2i). This analysis revealed several statistically significant relationships, of which the most significant pairs were DM3-D ($r = 0.49$, $p = 2.7 \times 10^{-4}$) and VL2p-VA6 ($r = -0.47$, $p = 4.9 \times 10^{-4}$) (Fig. 2i). In addition to DM3-D and VL2p-VA6, this analysis also revealed a significant probability for MIPergic LN co-innervation amongst several pairs of glomeruli responsive to ACV[25], such as VM2-DM1 ($r = 0.35$, $p = 0.01$), DM4-DM2 ($r = 0.31$, $p = 0.03$), and DP1m-DM1 ($r = 0.29$, $p = 0.04$). This suggests that the glomerular innervation patterns of individual MIPergic LN likely do not explain non-uniform MIPergic modulation of OSN (Fig. 1). However, it is plausible that differential modulation of OSN responses by MIP arises as a consequence of the non-uniform MIPergic LN pre-/postsynaptic sites across these glomeruli. Therefore, we sought to determine whether MIPergic LN input/output sites were heterogeneously distributed to any particular glomeruli throughout the AL.

## MIPergic LNs provide and receive broad input and output across the AL

We have shown that no individual MIPergic LN innervates the same set of glomeruli from animal-to-animal (Fig. 3a), but every glomerulus is innervated by at least one MIPergic LN across all animals (Fig. 3b). Therefore, we wondered whether significant differences in MIPergic LN input/output between glomeruli exist (Fig. 3c), which would explain the non-uniform effects of MIPergic modulation. To test this, we measured the density of MIPergic LN-expressed mCD8::GFP, anti-MIP immunoreactive punctae, and the synaptic polarity markers DenMark and synaptotagmin.eGFP (syt.eGFP)[56,57] in each glomerulus across many animals (Fig. 3d, e). We find that the density of each indicator varies across glomeruli but are stereotypic across samples (Fig. 3e and Supplementary Fig. 3f–i). The density of the output indicators (syt.eGFP and MIP-ir puncta) were statistically correlated, and nearly every indicator scaled with MIPergic LN intraglomerular cable density (Supplementary Fig. 3f–i). Even so, we find within-indicator voxel densities are generally evenly distributed across each glomerulus, suggesting MIPergic LN input and output are evenly distributed across the AL (Fig. 3e).

These puncta analyses afford the advantages of analyzing MIPergic LN synaptic polarity across many individuals of both sexes. However, light microscopy is limited by its inability to resolve fine structures such as axons/dendrites[58,59]. Therefore, we performed similar analyses on individual putative MIPergic LNs (putMIP LNs) within the most densely-reconstructed *Drosophila* central brain EM volume to-date, the hemibrain[60,61]. Additionally, EM-level analyses of putMIP LN connectivity have the added benefit of shedding light on what type(s) of neuron(s) and/or stimuli might generally promote MIP recruitment in AL processing. More specifically, EM analysis of putMIP LN connectivity allowed us to determine: (i) do certain glomeruli receive more input from putMIP LNs (and vice versa) than others? (ii) what neurons are upstream/downstream of putMIP LNs in each glomerulus? and, (iii) at which putMIP LN presynaptic terminals are vesicles associated with neuropeptides (dense core vesicles, or DCVs)[62,63] found? Thus, we first used several criteria to identify fully-reconstructed putMIP LNs, such as the candidates' principal identity, previous AL LN subtyping results[28], synaptic connectivity, and neuromorphic similarity to R32F10-GAL4 AL LNs (see Methods) (Fig. 3f). These stringent criteria resulted in the identification of 14 ideal candidates (Fig. 3f).

After identifying several optimal candidates, we tested whether any putMIP LNs have distinct dendritic and axonic compartments. If true, this would suggest putMIP LNs make region-specific input/output, as has been suggested for the "heterogeneous LNs" in the honeybee AL[64–67]. Synaptic flow centrality and axonal-dendritic segregation indices[68] reveal all putMIP LNs lack clearly separable input/

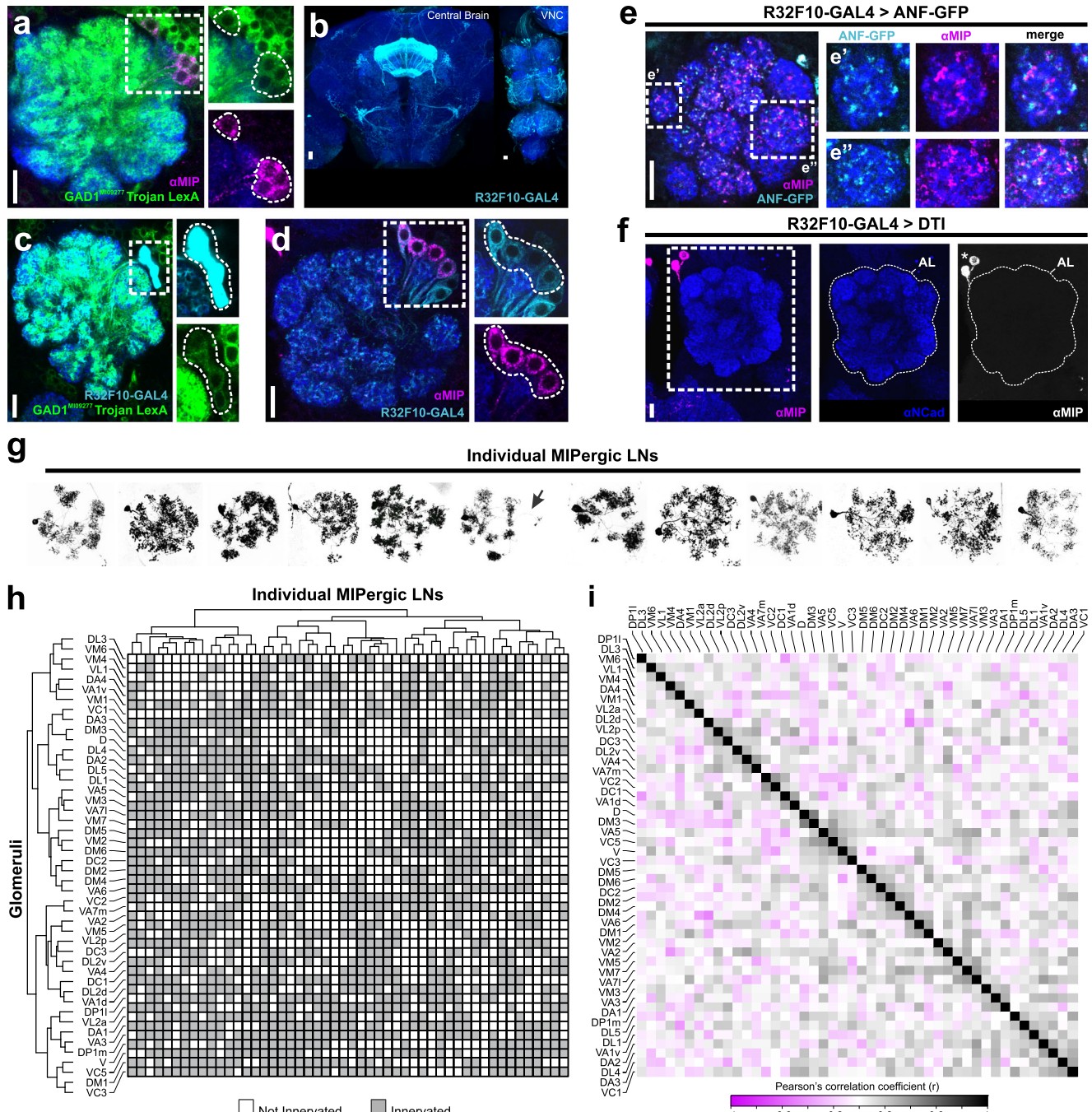

**Fig. 2 | Myoinhibitory peptide (MIP) is released by GABAergic patchy LNs in the AL. a** A protein-trap Trojan LexA driver for glutamic acid decarboxylase (GAD1), the rate-limiting enzyme for GABA, highlights all MIP immunoreactive (MIP-ir) neurons in the AL. Cell counts: n = 5 brains, 10 ALs. **b** R32F10-GAL4 expression in the central brain and ventral nerve cord (VNC). **c** All R32F10-GAL4 AL LNs colocalize with the GAD1 Trojan LexA protein-trap driver. Cell counts: n = 5 brains, 9 ALs. **d** All MIP-ir AL neurons (-8.7 ± 0.3 neurons) are highlighted by R32F10-GAL4, which labels -13.2 (±0.68) AL neurons in total. In addition to labeling all MIP-ir AL neurons, R32F10-GAL4 also labels -4.5 (±0.68) non-MIPergic GABAergic AL neurons. Cell counts: n = 5 brains, 9 ALs. **e** Representative image of GFP-tagged preproatrial natriuretic factor (ANF-GFP) expression in R32F10-GAL4 AL LNs. ANF-GFP accumulates with MIP-ir punctae in R32F10-GAL4 AL LN terminals, such as those in DM5 (e') and VA1d (e"). **f** Expression of a temperature-sensitive diphtheria toxin (DTI) in R32F10-GAL4

AL LNs abolishes all MIP-immunoreactivity in the AL (n = 10 brains, 20 ALs). Asterisks indicates the median bundle cluster of MIP-ir neurons outside of the AL, which remains intact when R32F10-GAL4 AL LNs are ablated. **g** Stochastic labeling of individual R32F10-GAL4 AL LNs reveals MIP is released by patchy LNs. Arrow = bilateral projection. **h** Glomerular innervation patterns of 50 individual MIPergic LNs organized by hierarchical clustering similarity. Each row represents the innervation pattern of a single clone, and each column represents a given glomerulus. Only the ipsilateral innervation patterns of the individual MIPergic AL LNs were included for analysis. (**i**) All pairwise correlations of MIPergic LN innervation patterns between AL glomeruli. Values correspond to the Pearson's correlation coefficient for each glomerulus pair. In all cases: neuropil was delineated by anti-DN-Cadherin staining; scale bars = 10 µm. Source data are provided as a Source Data file.

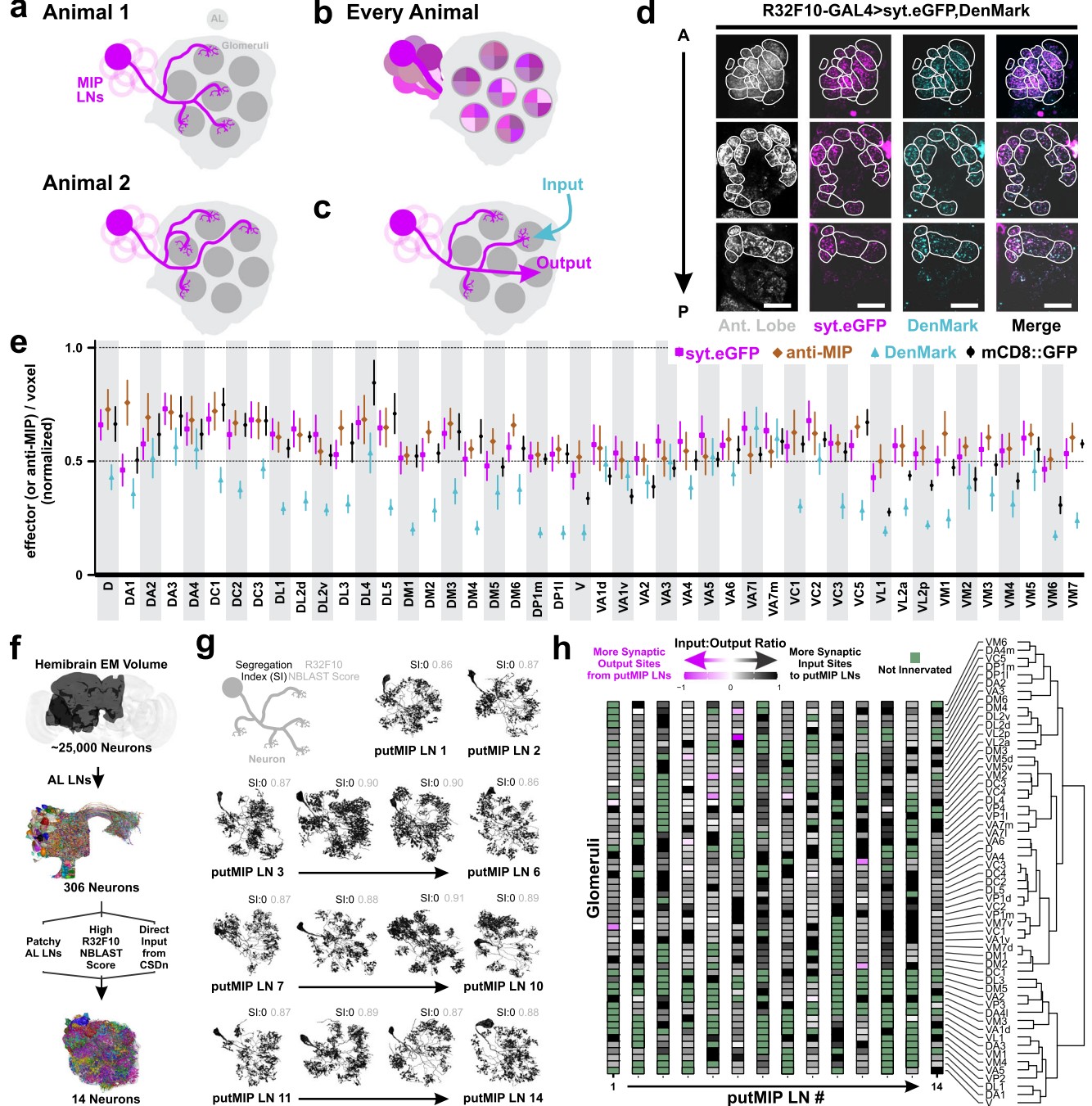

**Fig. 3 | MIPergic LN input and output sites throughout the entire AL.**
**a** Individual MIPergic LNs project to different glomeruli from animal-to-animal. **b** The MIPergic LN ensemble covers the entire AL across all animals. **c** Do MIPergic LNs receive input from particular sets of glomeruli? Are there particular sets of glomeruli subject to more/less MIPergic LN output than others? **d** Representative image of glomerular voxel density analysis. Here, MIPergic LNs express synapto-tagmin.eGFP (syt.eGFP; magenta) and DenMark (cyan) and their respective density is measured within each AL glomerulus (Ant. Lobe; gray). Glomeruli outlined in white. Scale bars = 10 μm. **e** syt.eGFP (magenta), DenMark (cyan), mCD8::GFP (black) and anti-MIP (brown) puncta density per voxel within each AL glomerulus. Each indicator is normalized to the highest value within that indicator. Data are represented as the mean ± SEM of each indicator's voxel density within a given

glomerulus. For each indicator, $n = 7$ (syt.eGFP), 7 (DenMark), 4 (mCD8::GFP), 4 (anti-MIP) brains. **f** Schematic representation of procedures used to identify ideal putative MIPergic LN (putMIP LN) candidates from the FlyEM FIB-SEM hemibrain connectome volume (see Methods). **g** putMIP LN mesh skeletons identified from the hemibrain EM volume. For each neuron, values in the upper right-hand corner are that neuron's synaptic segregation index (black) and R32F10-GAL4 NBLAST similarity score (gray). **h** putMIP LN intraglomerular input:output ratio across the AL. Each column represents a given glomerulus, and each row represents the input:output ratio of a single putMIP LN. Glomeruli not innervated by the given putMIP LN are green. Glomeruli are organized by hierarchical clustering similarity. Data only consider putMIP LN connections within the ipsilateral AL. Source data are provided as a Source Data file.

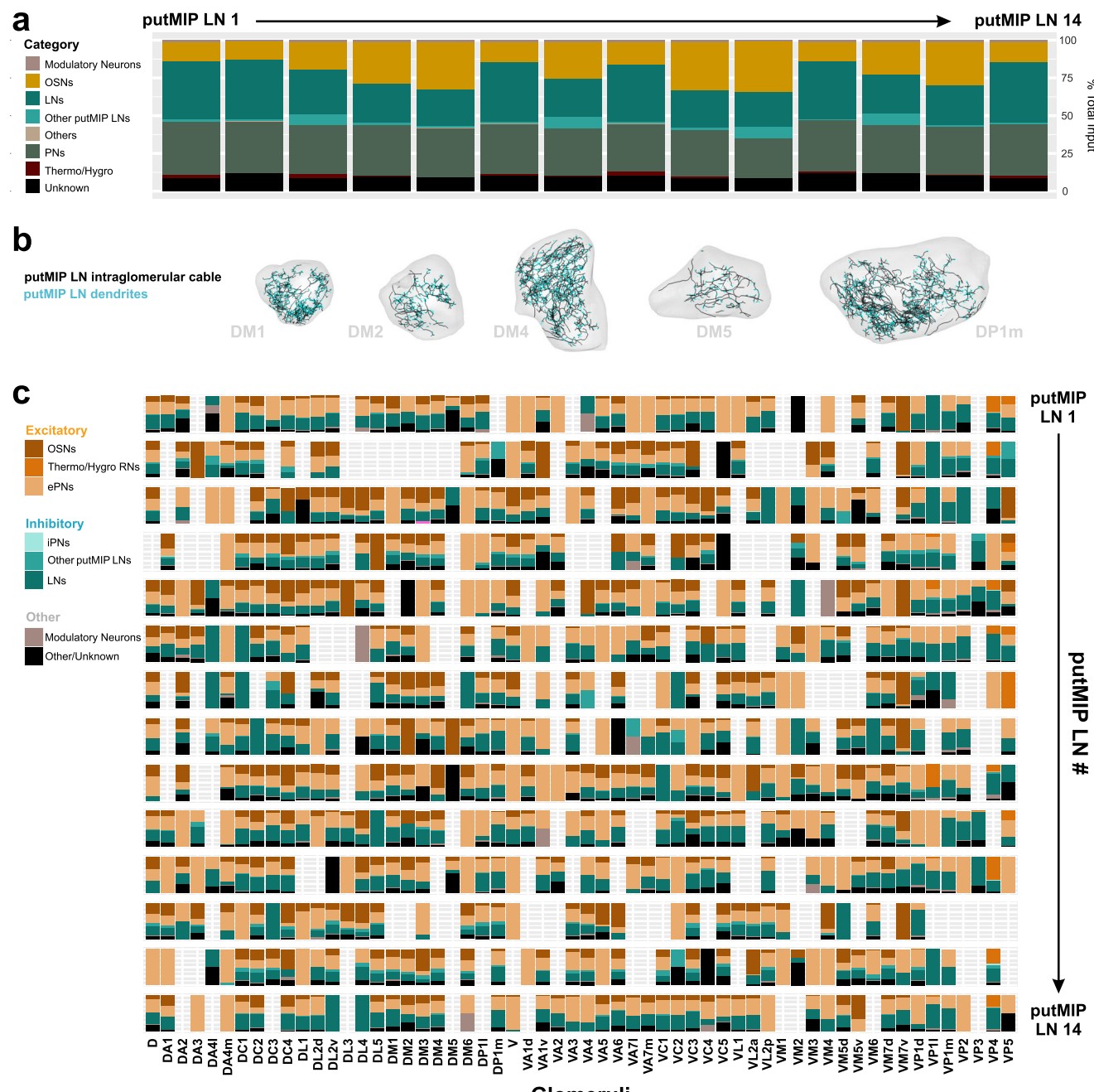

**Fig. 4 | Anatomical inputs to putMIP LNs and functional glomerular outputs from identified MIPergic LNs. a** putMIP LN upstream partners' demographics. Data are represented as a function of the total amount of input a putMIP LN receives from all categories. **b** Representative images of putMIP LN intraglomerular cable and dendrites plotted within their corresponding glomeruli. The amount of excitatory, inhibitory, and modulatory input each putMIP LN receives within every glomerulus as a function of the total amount of input a given putMIP LN receives within the glomerulus. **c** The amount of excitatory, inhibitory, and modulatory input each putMIP LN receives within every glomerulus, broken apart by presynaptic neuron identity, and represented as a function of the total amount of input a given putMIP LN receives within the glomerulus. Glomeruli with no bar graph are those that the given putMIP LN does not innervate. Source data are provided as a Source Data file.

output compartments (Fig. 3g). When we assess the ratio of input-to-output along a given putMIP LN's intraglomerular neurites, we find that the amount of input a given putMIP LN receives typically outnumbers the amount of putMIP LN output within any given glomerulus (Fig. 3h). To better understand how these inputs might drive MIPergic modulation, we assessed the general identity of all inputs a putMIP LN receives, as well as the class and transmitter type of each presynaptic input an intraglomerular putMIP LN arbor receives.

Generally, nearly half of all putMIP LNs receive more input from other LNs than other principal neuron categories (6/14 putMIP LNs;

~38-40% total input) (Fig. 4a). Similarly, just as many putMIP LNs receive the majority of their input from PNs than any other principal neuron category (6/14 putMIP LNs; ~31–33% total input) (Fig. 4a). Additionally, we found that within a given glomerulus putMIP LNs largely avoid one another, but do occasionally form synaptic connections (Fig. 4b and Supplementary Fig. 3j, k). When we refine these analyses by considering the putMIP LN's intraglomerular connectivity and the presynaptic partner's identity, we find the amount of excitatory or inhibitory input a given putMIP LN receives varies greatly across glomeruli. However, in every case, putMIP LNs generally receive

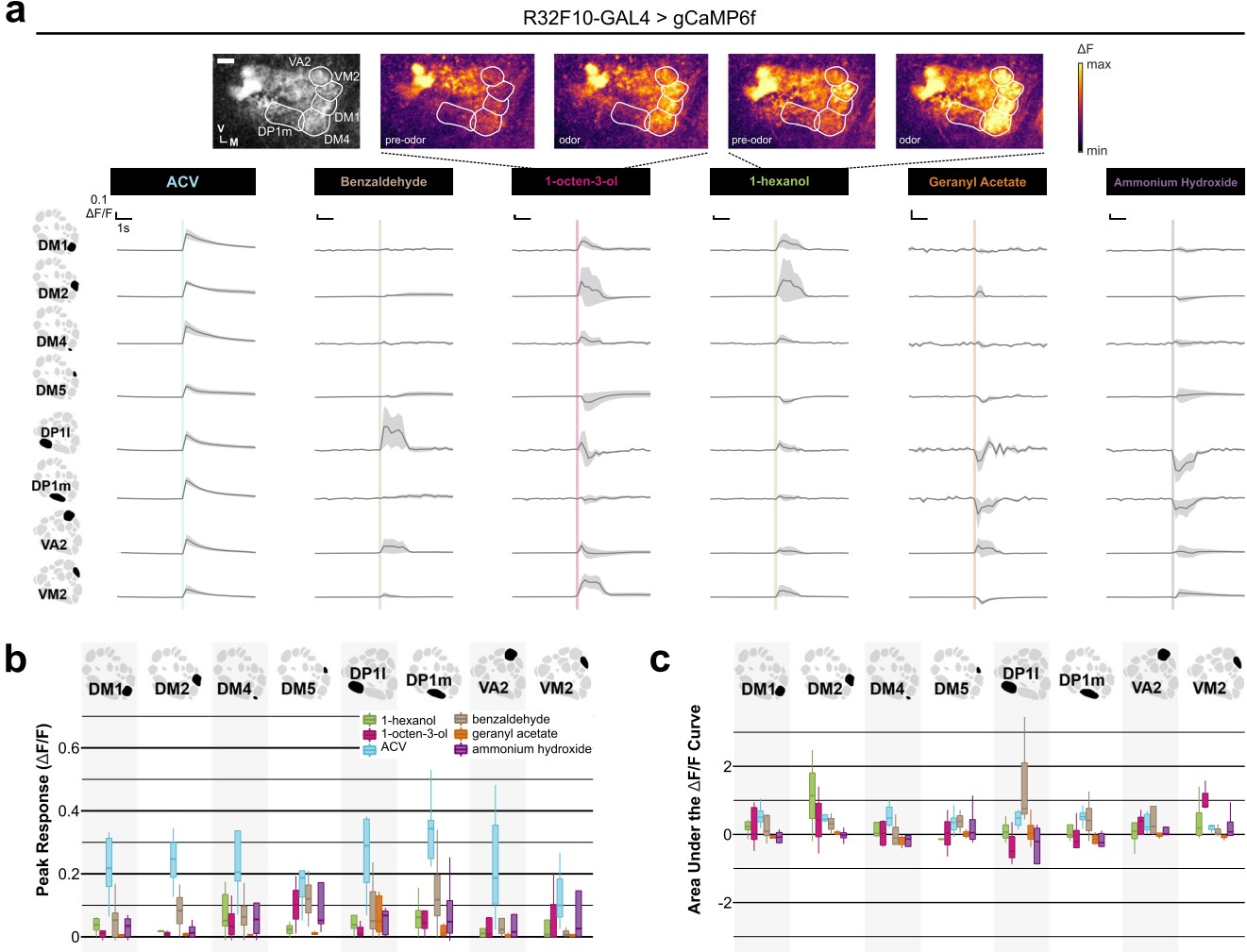

**Fig. 5 | MIPergic LNs are differentially activated by different odors. a** Odor-evoked responses of MIPergic LN neurites within several AL glomeruli (far left column). Scale bar = 10 μm. Odors tested were presented at $10^{-2}$ and include: apple cider vinegar (ACV), benzaldehyde, 1-octen-3-ol, 1-hexanol, geranyl acetate, and ammonium hydroxide. For ACV, $n = 3$ (VM2), 4 (DP1l), 5 (DM4 & VA2), 7 (DM2), 8 (DM5) and 10 animals (DM1 & DP1m). For benzaldehyde, $n = 3$ (DM2, DM5, DP1l, & VM2), 4 (DM4, DP1m, & VA2), and 5 animals (DM1). For 1-octen-3-ol, $n = 3$ (DM2, DM5, DP1l, & VM2), 4 (DM4, DP1m, & VA2), and 5 animals (DM1). For 1-hexanol, $n = 3$ (DM2 & DM5), 4 (DP1l & VM2), 5 (DM4, DP1m, & VA2), and 6 animals (DM1). For geranyl acetate, $n = 3$ (DM2 & DM5), 5 (DM4, DP1l, VA2, & VM2), and 6 animals (DM1 & DP1m). For ammonium hydroxide, $n = 4$ (DM2, DM5, & VM2), 5 (DM4, DP1l, &

VA2), and 6 animals (DM1 & DP1m). Data are presented as the mean (black line) ± SEM (gray shaded area). Vertical and horizontal scale bars = 0.1 ΔF/F & 1 s (respectively). Odor onset is indicated by the vertical lines running up each column of traces. **b** Peak response (ΔF/F) of MIPergic LN intraglomerular neurites from odor onset to -1 s after stimulus onset across all glomeruli tested for each stimulus. **c** Area under the ΔF/F curve (AUC) of MIPergic LN intraglomerular neurites across glomeruli for each stimulus. In all cases: Boxplots display the minimum, 25th-percentile, median, 75th-percentile, and maximum of the given data. Glomerular schematics were derived from an in vivo AL atlas[164]. Source data are provided as a Source Data file.

far more excitatory than inhibitory input within any given glomerulus (~65–93% of all glomeruli innervated by the given putMIP LN) (Fig. 4b). This suggests that MIPergic LN intraglomerular processes may be broadly activated by disparate odorants, which would suggest uniform release of MIP may occur in response to various odors. Therefore, we tested whether MIPergic LNs are broadly activated in vivo by chemically diverse odorants.

**MIPergic LNs generally display glomerulus-specific odor-evoked responses**

Synapse counts have been shown to strongly predict functional output strength in neurons within other systems, including other *Drosophila* AL neurons[28,69–74]. Our connectomic analyses intraglomerular putMIP LN arbors generally receive mostly excitatory input within a given glomerulus (Fig. 4b), which would imply that MIPergic LNs are broadly activated regardless of odor identity. This would be consistent with

previous characterizations of large ensembles of GABAergic AL LNs odor-evoked GCaMP responses that have observed odor-invariant activation across nearly all glomeruli[54,75]. Therefore, we recorded the in vivo odor-evoked responses of MIPergic LN intraglomerular neurites to a panel of chemically diverse odorants across multiple glomeruli (Fig. 5a). Moreover, we chose to image within glomeruli whose cognate PNs respond to at least one odorant in our test panel to better understand what role the MIPergic LNs may play in the OSN-to-PN information transfer. For example, benzaldehyde and geranyl acetate each evoke responses in DM1, DM4, DP1l, VA2, and VM2 PNs[37,76–78].

In contrast to other GABAergic AL LNs[54,75], we find MIPergic LNs generally display glomerulus-specific responses to all test odors (Fig. 5a). For example, MIPergic LN neurites within VM2 and DP1m—two glomeruli visible at the same imaging depth—are simultaneously activated and inhibited by 1-octen-3-ol, respectively (Fig. 5a–c). This same odor drove post-excitatory depression in MIPergic LN neurites

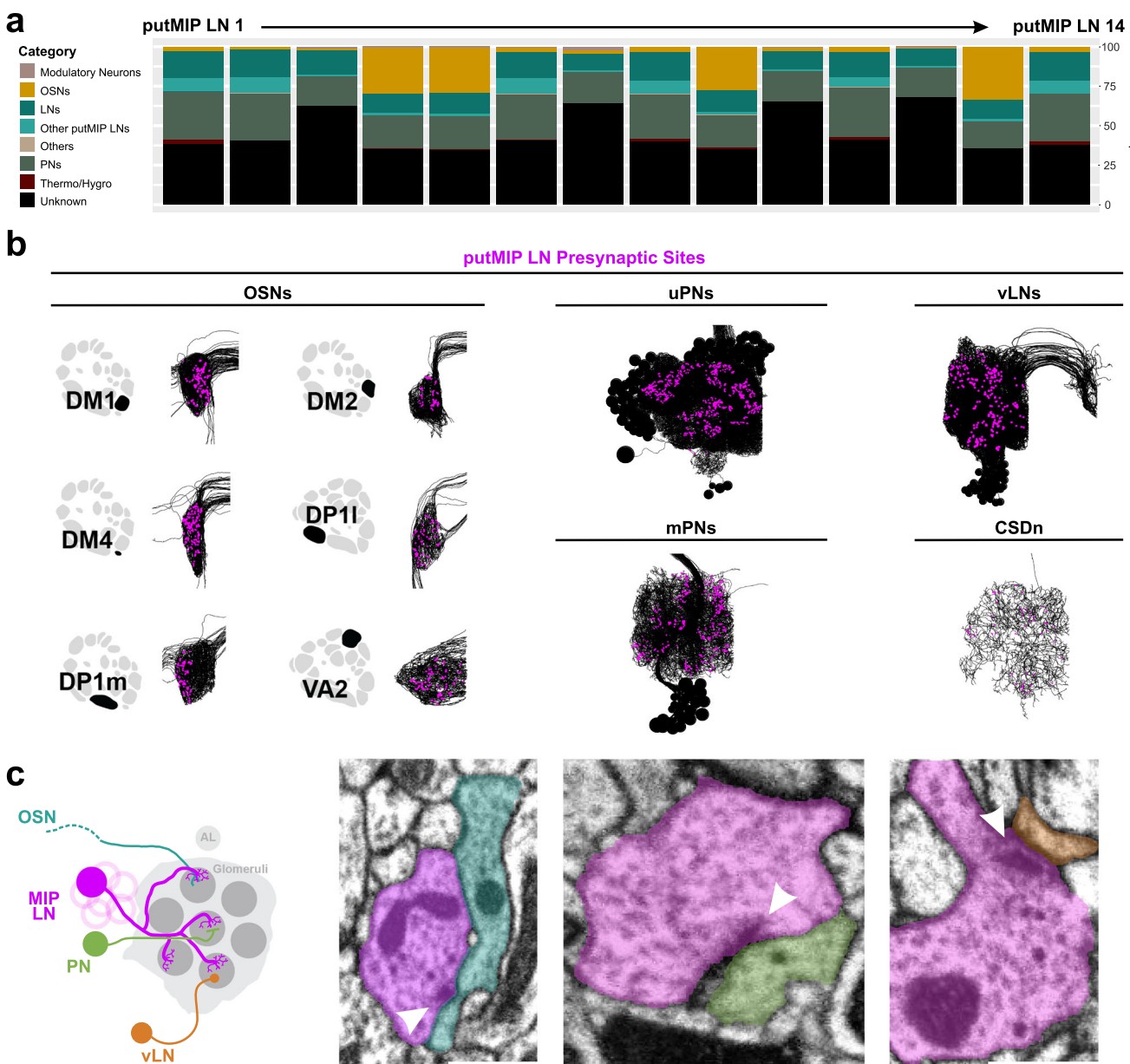

**Fig. 6 | Postsynaptic targets of each putMIP LN and representative putMIP LN presynaptic terminals with dense core vesicles (DCVs). a** Demographics of all putMIP LN postsynaptic targets by neuron type. Data are represented as a function of the total amount of output a putMIP LN sends to all categories. **b** Representative putMIP LN postsynaptic partner skeletons (black) with their respective putMIP LN presynaptic locations (magenta). Glomerular schematics were derived from an in vivo AL atlas[164]. **c** Representative instances where DCVs in the putMIP LN presynaptic terminal (n = 14 putMIP LNs). From left to right: DCVs are in putMIP LN presynaptic terminals upstream of OSNs (cyan), PNs (green), and ventral LNs (vLN; orange). In all cases: white arrowheads indicate the putMIP LN's presynaptic site; scale bars = 500 nm. Source data are provided as a Source Data file.

within DP1l and VA2 (Fig. 5a–c). However, in many cases MIPergic LN intraglomerular neurites did not respond to the given odorant (Fig. 5a–c). This shows that, unlike other GABAergic AL LNs which are broadly activated in response to similar stimuli[33,54,75], MIPergic LN intraglomerular processes are differentially activated by different odors. However, we acknowledge that these odor-evoked responses do not necessarily reflect MIP release itself, which we were unable to test for reasons described below (see Discussion).

Notably, ACV elicited robust activation of MIPergic LN intraglomerular processes across all identifiable glomeruli (Fig. 5a–c), including those glomeruli we found MIP non-uniformly modulates (Fig. 1). This finding, together with our earlier results (Figs. 2–5), suggests that the non-uniform effects of MIP on olfactory input likely do not arise

from the presynaptic MIP-releasing neurons themselves. Instead, these results suggest the non-uniform effects of MIP on olfactory input are an emergent property of either (i) MIPergic LN postsynaptic targets, and/or (ii) differential SPR expression across the AL.

### MIPergic LN downstream partners and widespread SPR expression within the AL

To determine the AL principal neurons likely targeted by MIPergic LNs, we first assessed the general output demographics for each putMIP LN (Fig. 6a, b). Of the AL principal neuron types, most putMIP LNs chiefly target PNs (71% of putMIP LNs; ~19–31% of putMIP LN total output) (Fig. 6a). The remaining minority of putMIP LNs are chiefly presynaptic to OSNs (29% of putMIP LNs; ~27–33% of putMIP LN total output)

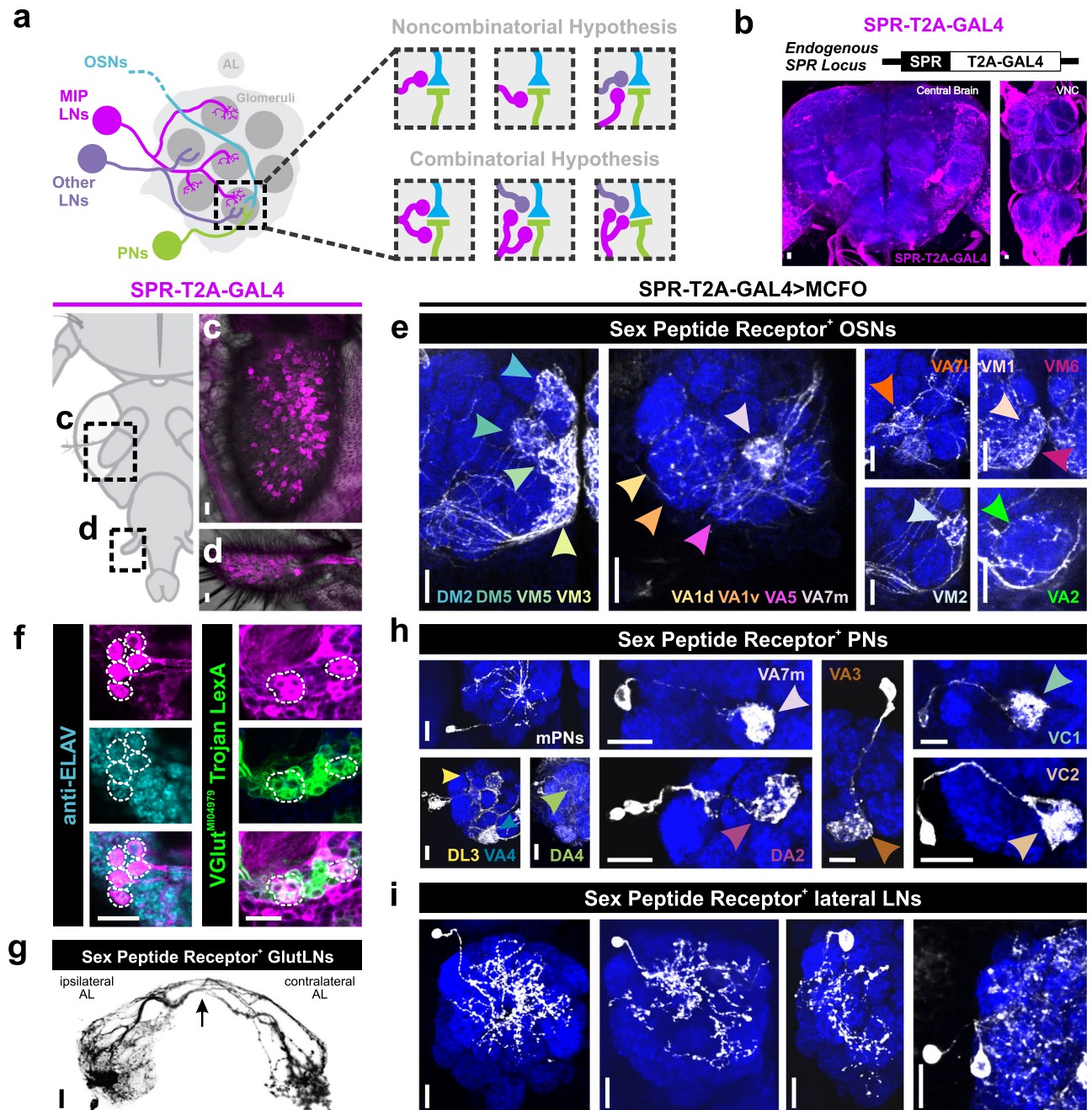

**Fig. 7 | Widespread sex peptide receptor (SPR) expression throughout the AL.**
**a** MIPergic LNs (magenta) form synaptic connections with all principal neuron types in the AL; OSNs (cyan), PNs (green), and other LNs (purple). Therefore, within a single glomerulus, MIPergic modulation might target any one of these neuron types (Non-combinatorial Hypothesis), or multiple neuron types (Combinatorial Hypothesis). **b** SPR expression (magenta) revealed through a CRISPR/Cas9 T2A-GAL4 insertion in the SPR-coding intron (n = 5 animals). **c, d** SPR-T2A-GAL4 expression in OSNs in the 3rd-antennal segment and maxillary palp (n = 17 and 18 animals, respectively). **e** SPR-T2A-GAL4 stochastic labeling experiments where the antennal nerve remains intact reveals SPR-expressing OSNs project to: DM2, DM5,

VM5v, VM5d, VM3, VA1d, VA1v, VA5, VA7m, VA7l, VM1, VM6, VM2, and VA2 (n = 31 animals). **f** Several SPR-T2A-GAL4 neurons are immunopositive for the proneural marker ELAV (cyan), a subset of which colocalize with VGlut[MI04979] Trojan LexA (green) (n = 23 animals). **g** SPR-T2A-GAL4 stochastic labeling reveals several bilaterally-projecting ventral glutamatergic LNs (GlutLNs). Arrow = bilateral projection. **h** Several PNs are highlighted via SPR-T2A-GAL4 stochastic labeling, which include ~3 multiglomerular PNs and lateral and anterodorsal PNs that project to: DA2, DA4, DL3, VA3, VA4, VA7m, VC1, and VC2. (i) Approximately five lateral LNs are identified through SPR-T2A-GAL4 stochastic labeling. For **h** and **i**, n = 31 animals. In all cases: neuropil was delineated by anti-DN-Cadherin staining; scale bars = 10 μm.

(Fig. 6a). Since AL PNs express GABA_A and GABA_B receptors[27], these results would suggest that MIPergic LNs may provide fast- and slow-acting inhibition across the AL, perhaps as a means to normalize PN odor-evoked responses. To determine which of these downstream partners (Fig. 6b) are likely targeted by MIPergic modulation, we determined which postsynaptic partners were downstream of putMIP LN terminals where dense core vesicles (DCVs) are observable (Fig. 6c).

We observed several instances where DCVs could be found in putMIP LN terminals presynaptic to OSNs, PNs, and ventral LNs (Fig. 6c). However, MIPergic LNs could also release other neuropeptides, so the presence of DCVs in MIPergic LN presynaptic terminals does not necessarily mean the downstream neuron is modulated by MIP. Moreover, putMIP LN EM analyses indicate several AL principal neuron types are plausible targets for MIPergic modulation (Figs. 3, 6, and 7a).

To determine which downstream partners are subject to MIPergic modulation, we identified the AL neurons that express MIP's cognate receptor, the $G_{\alpha\text{-i}}/G_{\alpha\text{-o}}$-coupled sex peptide receptor (SPR)[79–82]. To do so, we used a CRISPR/Cas9-mediated T2A-GAL4 insertion within the endogenous SPR locus to enable GAL4 expression within SPR-expressing cells[83] (Fig. 7b).

In *Drosophila*, OSN somata are located within the third-antennal segment and maxillary palp[84,85]. We find $208.9 \pm 11.89$ ($n = 17$ animals, 30 antennae) and $63.42 \pm 4.31$ ($n = 18$ animals, 31 maxillary palps) SPR-T2A-GAL4⁺ neurons in the third-antennal segment and the maxillary palp, respectively (Fig. 7c, d). As there are ~945 and ~113 OSNs in the antennae and maxillary palps, respectively[86], this would suggest that ~22% of antennal OSNs and ~56% of maxillary palp OSNs express SPR. The number of SPR-T2A-GAL4⁺ neurons in either appendage do not significantly differ based on the animal's sex or mating status (antennae: $p = 0.107$, one-way ANOVA; maxillary palps: $p = 0.559$, Kruskal-Wallis test). However, this does not discount differences in the level of SPR expression within these neurons based on the animal's sex or mating status. Through stochastic labeling experiments where the antennal nerve is left attached to the brain, we found OSN fibers that innervate many distinct glomeruli, including several ACV-responsive OSNs (Fig. 7e and Supplementary Data 1). Interestingly, we found SPR-T2A-GAL4 expression in afferents belonging to every sensory modality (Supplementary Fig. 4), which suggests MIPergic modulation of sensory afferents may be a fundamental feature in *Drosophila*.

Within the brain, we find consistent colocalization of SPR-T2A-GAL4 and the proneural gene *embryonic lethal abnormal vision* (ELAV) (Fig. 7f) and the glial marker *reverse polarity* (REPO) (Supplementary Fig. 4). Through intersectional genetics and stochastic labeling, we find that these ELAV-positive neurons are composed of: $4.89 \pm 0.21$ ($n = 23$ brains, 44 ALs) SPR-expressing ventral glutamatergic LNs (GlutLNs) (Fig. 7f, g), uniglomerular PNs (Fig. 7h and Supplementary Data 1), and several lateral LNs (Fig. 7i). In agreement with these results, we find similar neuron types using another SPR driver (SPR-GAL4::VP16)[87] (Supplementary Fig. 5), several publicly available scRNA-seq datasets[88–90] (Supplementary Fig. 6), and a novel SPR^{MI13553}-T2A-LexA::QFAD driver (Supplementary Fig. 7). Notably, SPR-T2A-GAL4⁺ neurons were consistent across all animals—a stark contrast to the animal-to-animal differences in individual MIPergic LN morphology previously observed (Fig. 2).

### Differential SPR expression across glomeruli enables non-uniform MIPergic modulation of olfactory input

To test the necessity of direct MIP-SPR signaling on modulation of OSN odor-evoked responses, we repeated our earlier experiments (see Fig. 1), but used RNA interference (RNAi) to knockdown SPR specifically within OSNs (Fig. 8a, b). Moreover, this SPR-RNAi has been used to effectively knockdown SPR in OSNs previously[24], and abolishes SPR immunoreactivity in the *Drosophila* CNS when expressed pan-neuronally[91]. We find that SPR knockdown abolishes the MIP-induced decrease in the odor-evoked responses of DM2 and DM5 OSNs (Fig. 8c, d). This result is consistent with SPR-expression in DM2 and DM5 OSNs (Fig. 7e), and suggests MIP directly decreases the odor-evoked responses of DM2 and DM5 OSNs. In contrast, SPR knockdown in DM1 and DM4 OSNs does not prevent their responses from increasing after peptide application (Fig. 8c, d). Since we did not observe SPR-expression in DM1 and DM4 OSNs (Fig. 7), and SPR knockdown in these OSNs does not abolish the MIP-induced increase in their responses (Fig. 8c, d), our results suggest MIP acts polysynaptically to disinhibit (thus, increasing) DM1 and DM4 OSN odor-evoked responses (Fig. 8e).

### Discussion

Our data reveals the circuit topology that enables a single neuropeptide, acting through a single receptor, to differentially modulate olfactory processing. We show that pharmacological application of MIP elicits non-uniform and complex effects on olfactory input to the *Drosophila* primary olfactory center. Here, MIP reduces the responses of OSNs in some glomeruli, and simultaneously enhances the responses of OSNs in other glomeruli (Fig. 1). We show that the non-uniform effects of MIP on olfactory input is likely not an emergent property of the identity, structure, and/or connectivity of the MIP-releasing neurons, themselves. Instead, we find that differential SPR expression within distinct glomeruli enables MIP to non-uniformly modulate olfactory input across olfactory channels.

We found that individual MIPergic LNs innervate a different repertoire of glomeruli across animals and do not preferentially innervate any one glomerulus over others (Fig. 2 and Supplementary Fig. 2). These findings are consistent with earlier reports wherein patchy AL LNs were first generally described[32]. But, what factor(s) give rise to the tremendous flexibility within this single morphological subtype? One explanation might be that MIPergic LN morphological idiosyncrasy is a byproduct of experience during development. However, OSN removal in the adult does not disrupt the animal-to-animal variability of patchy LNs[32]. To the best of our knowledge, a single locus (e.g., environmental experience or heritable trait) that would support animal-to-animal variation in patchy LNs has not been identified.

Another explanation for animal-to-animal differences in individual MIPergic LN morphology is that it may not matter which individual MIPergic LN forms synapses with which downstream target, as long as all of the MIPergic LN downstream targets are met. Every nervous system is the byproduct of the adaptive pressures demanded by the animal's niche; a place that can continually change in seemingly unpredictable ways. Therefore, a developmental parameter space may exist, wherein just enough genetic idiosyncrasy is allowed to help prevent extinction in the face of environmental perturbations. The breadth of this developmental parameter space (or the degree of variability from the median) would be defined by many generations of selective pressures, wherein subtle changes in genetic idiosyncrasies might equally result in winners and losers. As a consequence of these genetic idiosyncrasies, phenotypic variability in a given developmental program would inevitably accumulate, resulting in the observed animal-to-animal variability in neuronal features (e.g., morphology, ion channel distribution, etc.). Consistent with this idea, animal-to-animal variations in neural circuitry have been noted in grasshoppers[92], crabs[93–97], lobsters[98,99], flies[32,100,101], and rats[102]. Moreover, inter-animal variations in neuronal architecture are one of several features implicated in inter-animal behavioral variations[101,103–107]. However, despite this variability, overall neuronal circuit functions persist including consistent MIPergic LN synaptic polarity marker density (Fig. 3), MIPergic LN within-odor responses (Fig. 5), MIP-induced decreases in DM2 and DM5 OSN responses across animals (Fig. 6), and SPR expression (Fig. 7). Moreover, several positive and negative correlations exist for pairs of glomeruli innervated by single MIPergic LNs, such as the significant probability for MIPergic LN co-innervation in ACV-responsive glomeruli (Fig. 2i). Together, these results suggest that the morphology of an individual MIPergic LNs can differ from animal-to-animal, as long as the right combinations of downstream targets (e.g., ACV-responsive neurons) are met by the ensemble.

We have shown that a small ensemble (~5% of all AL LNs) of GABAergic AL LNs are the sole source of MIP in the *Drosophila* AL (Fig. 2 and Supplementary Fig. 1). This implies that MIPergic LNs have the capacity to adjust AL olfactory processing through both GABA- and MIP-release. Previous works found that the iono- and metabotropic GABA receptors are expressed amongst all AL principal neuron types[27,29,108], and we show SPR is similarly expressed by members of every AL principal neuron type (Fig. 7). Therefore, MIPergic LN activation could plausibly cause both fast-acting and long-lasting inhibition in the same and/or disparate downstream target. Moreover, MIPergic LN-derived GABA and MIP may simultaneously act on the same downstream target(s) to synergistically modulate their activity to

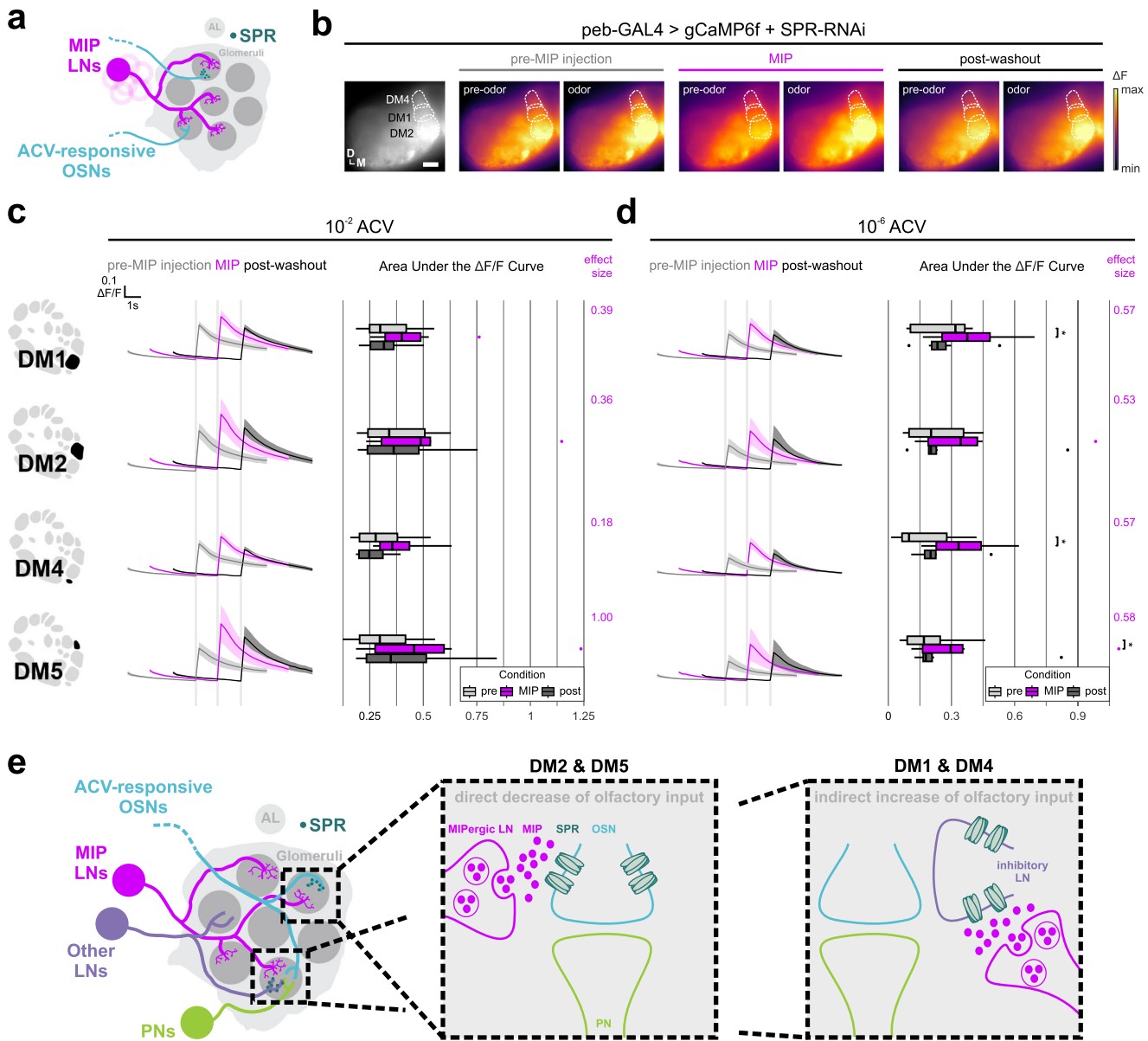

**Fig. 8 | SPR knockdown in OSNs reveals heterogeneous SPR expression across glomeruli enables non-uniform MIPergic modulation of OSN ACV responses.**
**a** Individual MIPergic LNs (magenta) significantly co-innervate several ACV-responsive glomeruli (cyan). Moreover, ACV-responsive OSNs (cyan) form synaptic connections with MIPergic LNs (magenta) and express the MIP receptor, SPR (turquoise). **b** Representative pseudocolored heatmaps of OSN GCaMP responses (ΔF) before and during odor presentation in several test glomeruli (white dotted outlines) of animals where SPR is knocked down. In each case, each odor presentation heatmap pair is grouped by stage of MIP pharmacological application. Scale bar = 10 μm. **c, d** SPR knockdown in OSNs abolishes MIP-induced decrease in DM2 and DM5 OSN responses (DM2 $10^{-2}$: $p = 0.136$, RM one-way ANOVA; $10^{-6}$: $p = 0.063$, pre-MIP vs. MIP & $p = 0.688$, pre-MIP vs. post-washout; Holm-adjusted Wilcoxon signed-rank test; $n = 6$; DM5 $10^{-2}$: $p = 0.135$, RM one-way ANOVA; $10^{-6}$: $p = 0.063$, pre-MIP vs. MIP & $p = 0.313$, pre-MIP vs. post-washout; Holm-adjusted Wilcoxon signed-rank test; $n = 6$). In contrast, SPR knockdown in OSNs does not

abolish MIP-induced increases in DM1 and DM4 OSN responses (DM1: $p = 0.031$, pre-MIP vs. MIP AUC; Holm-adjusted Wilcoxon signed-rank test; $n = 7$; DM4: $p = 0.031$, pre-MIP vs. MIP AUC; Holm-adjusted Wilcoxon signed-rank test; $n = 7$). In all cases: Data are presented as the mean (darker center line) ± SEM (lighter shaded area). Vertical and horizontal scale bars = 0.1 ΔF/F & one second (respectively). Odor onset is indicated by vertical lines running up each column of traces. Boxplots display the minimum, 25th-percentile, median, 75th-percentile, and maximum of the given data. Statistical measures of effect size (either Kendall's $W$ or Cohen's $d$) are provided to the right of each set of AUC boxplots. All statistical tests were two-tailed. Glomerular schematics were derived from an in vivo AL atlas[164]. Source data are provided as a Source Data file. **e** Conceptual model of differential MIPergic modulation of OSN responses across multiple AL glomeruli. Our data suggests that the MIPergic LNs are the sole source of MIP to the AL, where MIP acts to directly decrease DM2 and DM5 OSN responses. Our data also suggest that MIP acts to indirectly increase DM1 and DM4 OSN responses, likely through disinhibition.

have a greater effect than either modulator alone might achieve. Alternatively, MIPergic LNs might primarily use GABA throughout the course of ongoing network activity, and use MIP only under special circumstances (see example below).

We attempted to parse the contribution of GABA and MIP over the course of our investigation. We conceived an experimental paradigm

in which MIPergic LNs would be artificially activated chemogenetically via P2X2 misexpression[109] and ATP injection, while we recorded OSN GCaMP responses before and after MIPergic LN activation. The results of these experiments would then be compared to similar experiments wherein MIPergic LN GAD1 expression would be knocked down as a means of resolving the contribution of MIPergic LN-derived GABA.

However, both approaches require prior knowledge of what level of artificial activation is necessary to mobilize MIPergic LN dense core vesicles (DCVs), and thus MIPergic LN-derived MIP release.

To resolve the minimal strength of artificial activation necessary to mobilize MIPergic LN DCVs, we artificially activated MIPergic LNs (described above), while simultaneously recording DCV changes via either ANF-GFP[48] or the neuropeptide release sensor NPPR-ANP-GCaMP6s[110]. However, we were unable to detect any change in either indicator even when we injected 100 mM ATP—a concentration 10x-greater than what is necessary to activate other AL LNs[111]. As a consequence, it remains infeasible to simply artificially activate MIPergic LNs, while measuring a downstream neurons' responses, and accurately attribute changes in the downstream neuron MIP released from MIPergic LNs.

Co-transmitters often increase the computational capacity of the neuron they are released from, and the plasticity of the networks in which they act[112–115]. Therefore, it is likely that future investigations using novel technology may find that the MIPergic LNs play a far greater role in AL processing than what is resolved here. As detailed above, each co-transmitter may confer unique and substantial contributions to the overall role these interneurons have in AL processing. As such, this aspect of MIPergic LN function is an important and remaining question that should be addressed by future investigations.

Generally, multiple glomeruli are activated by any given odorant[25,53–55,116]. However, optogenetics can be used to selectively activate individual glomeruli in a manner similar to their odor-evoked responses to evaluate the behavioral contribution of individual glomeruli[117]. Such experiments reveal that DM1 and DM2 co-activation do not summate, and co-stimulation of both glomeruli produces a behavioral response that resembles DM1-only activation[117]. Based on this, the existence of an antagonistic relationship between DM1 and DM2 was proposed, wherein co-stimulation reduces the efficacy of either or both glomeruli[117]. We find MIP indirectly increases DM1 and directly decreases DM2 OSN responses (Fig. 8). Therefore, MIP-SPR signaling in DM1 and DM2 may act as a homeostat such that coactivation of each glomerulus never produces a behavioral response greater than the DM1-only activation response. This buffer would be advantageous for preventing saturation at the downstream neurons that receives convergent input from these glomeruli[28,70,118,119].

MIP-SPR signaling has been implicated in several behavioral state switches[23,24]. Notably, abolishing MIP release by inactivating all MIPergic neurons, or using a MIP-genetic null mutation, increases the animal's drive for food-derived odors[23]. Moreover, DM2 OSN firing rate increases when all MIPergic neurons are inactivated[23]. In contrast, increasing the activity of all MIPergic neurons decreases attraction toward food-odors, to the extent of eliciting odor-induced aversion[23]. Together, these behavioral results suggest MIP-SPR signaling can affect the sensitivity to food-associated odors and drive to search for food. In accordance with these observations, we found that individual MIPergic LNs significantly co-innervate several food-odor associated glomeruli (Fig. 2) and neurons from several of these glomeruli express SPR (Fig. 7). Most strikingly, we find that MIP directly acts on DM2 OSNs to decrease their odor-evoked responses (Fig. 8). Furthermore, we show that the MIP-induced decrease in DM2 responses occurs in a stimulus-concentration independent manner (Figs. 1 and 8). Altogether, these results point to a probable role for MIPergic LN-derived MIP signaling to adjust olfactory processing, likely while other MIPergic neurons adjust other sensory/motor elements, in accordance with satiety homeostasis drives. However, this role is likely only one of many that the MIPergic LNs play in AL processing as they also release GABA, and form reciprocal connections with neurons outside of the SPR-expressing neurons (Figs. 3 and 4).

Peptidergic modulation can be as simple as a single neuropeptide modulating motor output in the stick insect locomotor system[120], or as complex as the ~37 neuropeptide families acting within the cortex[121].

Our data highlight how even a seemingly simple case, a single neuropeptide acting through a single receptor, can have complex consequences on network processing by acting non-uniformly within different components of the overall network. As neuropeptide functions are often deeply conserved, and as the actions of neuropeptides begin to come into focus, similar instances of complex and non-uniform peptidergic modulation will likely appear across disparate taxa and modalities.

## Methods

### Fly husbandry, genotypes, and subject details
A complete table of each animal's genotype used for each experiment are included in Supplementary Data 1. Information on parental stock origins and relevant identifiers are provided in Table 1. Unless otherwise noted, flies were reared on standard cornmeal and molasses media at 24 °C and under a 12:12 light:dark cycle. Equal numbers of male and female animals were used when possible, excluding live-imaging experiments which used only females. For mating status comparisons: 1) virgin females denotes females that were meconium-positive upon collection, 2) non-virgin females were housed with males until processing for immunohistochemistry, and 3) flies were age-matched and kept on the similar media until processed for immunohistochemistry.

### Immunohistochemistry and imaging
All immunohistochemistry was performed generally as previously described[122]. Briefly, samples were dissected, fixed in 4% paraformaldehyde, then washed with phosphate buffered saline with 0.5% Triton-X 100 (PBST) several times before taking samples through an ascending-descending ethanol wash series, then blocking in 4% IgG-free BSA (Jackson Immunoresearch; Cat#001-000-162). Samples were then incubated in primary antibody (Table 1) diluted in blocking solution and 5 mM sodium azide. Following primary antibody incubation samples were washed with PBST, blocked, and incubated in secondary antibody diluted in blocking solution and 5 mM sodium azide. Finally, samples were washed, cleared using an ascending glycerol series (40%, 60%, 80%), and mounted on well slides in Vectashield® (Vector Laboratories, Burlingame, CA; Cat#H-1200). Images were collected and analyzed as previously described[122] with VAA3D[123] and FluoRender[124], apart from those captured with a 40x/1.25 Silicone UPlanSApo Olympus objective. In some cases, image brightness and contrast were manually adjusted in CorelDRAW 2021 (Corel Corp., Ottawa, Canada). In the case of Fig. 7h, i, the FluoRender brush tool[124] was used to select and segment the relevant neuron(s) if they were obscured by the primary neurites of other cell(s) that occlude the cells-of-interest. All original light microscopy confocal scans used to create the figure panels are available at https://zenodo.org/10.5281/zenodo.8127341.

### Single LN clone induction and glomerular innervation analyses
Single LN clones were induced through the MultiColor Flip Out (MCFO) method[49]. Flies carrying the MCFO cassettes, Flp-recombinase, and GAL4 driver were raised under normal conditions (see above) until heat shock. Adult flies were heat-shocked in a 37 °C water bath for 12–25 min and returned to normal conditions for ~2–3 days before processing for immunohistochemistry. We chose to analyze the innervation patterns of 50 individual MIPergic LNs based on a statistical probability theorem termed, the coupon collector problem[125]. For our purposes, this meant we needed to sample 43 individual LNs to ensure we sampled each of the ~13 LNs highlighted by R32F10-GAL4 (Fig. 1b–d). We chose to analyze more than the minimal number as determined by this theorem as an additional preemptive measure to ensure the ~8 MIPergic AL LNs were sampled. Apart from VA1v, glomeruli were defined according to previously published AL maps[126,127]. Glomerulus names were later updated according to recent naming conventions[28]. Neuropil were labeled using anti-DN-cadherin or anti-

**Table 1 | Sources and identifiers for all key reagents and resources used in this present study**

| Reagent/Resource | Source | Identifier |
|---|---|---|
| **Antibodies** | | |
| Rabbit anti-RFP | Rockland | Catalog #: 600-401-379; RRID: AB_2209751 |
| Rabbit anti-DsRed | Clontech | Catalog #: 632496; RRID: AB_10013483 |
| Rat anti-DN-Cadherin | DSHB, University of Iowa | Catalog #: DN-Ex#8; RRID: AB_528121 |
| Rabbit anti-GFP | ThermoFisher Scientific, CA | Catalog #: A-11122; RRID: AB_221569 |
| Chicken anti-GFP | Abcam | Catalog #: ab13970; RRID: AB_300798 |
| Rabbit anti-hemagglutinin | Cell Signaling Technology | Catalog #: 3724; RRID: AB_1549585 |
| Mouse anti-V5-Tag::DyLight550 | BioRad (formerly AbD Serotec) | Catalog #: MCA1360D550GA; RRID: AB_2687576 |
| Rat anti-FLAG | Novus Bio | Catalog #: NBP1-06712SS; RRID: AB_1625982 |
| Mouse anti-bruchpilot | DSHB, University of Iowa | Catalog #: nc82; RRID: AB_2314866 |
| Rabbit anti-myoinhibitory peptide (MIP) | Manfred Eckert (gift from Christian Wegener) | RRID: AB_2314803 |
| Rat anti-embryonic lethal abnormal vision (ELAV) | DSHB, University of Iowa | RRID: AB_528218 |
| Mouse anti-reversed polarity (REPO) | DSHB, University of Iowa | RRID: AB_528448 |
| Goat anti-rabbit AlexaFluor 488 | ThermoFisher Scientific, CA | Catalog #: A-11008; RRID: AB_143165 |
| Donkey anti-chicken AlexaFluor 488 | Jackson ImmunoResearch Laboratories, Inc. | Catalog #: 703-545-155; RRID: AB_2340375 |
| Donkey anti-rabbit AlexaFluor 546 | ThermoFisher Scientific, CA | Catalog #: A-10040; RRID: AB_2534016 |
| Goat anti-mouse AlexaFluor 546 | ThermoFisher Scientific, CA | Catalog #: A-11030; RRID: AB_2534089 |
| Goat anti-rabbit AlexaFluor 633 | ThermoFisher Scientific, CA | Catalog #: A-21070; RRID: AB_2535731 |
| Goat anti-mouse AlexaFluor 633 | ThermoFisher Scientific, CA | Catalog #: A-21050; RRID: AB_2535718 |
| Donkey anti-rat AlexaFluor 647 | Abcam | Catalog #: ab150155 |
| **Odors** | | |
| Paraffin oil | J.T. Baker, VWR | CAS #: 8012-95-1 |
| Apple cider vinegar (ACV) | Heinz | N/A |
| 2-heptanone | Millipore-Sigma | Catalog #: 537683; CAS #: 110-43-0 |
| 1-hexanol | Millipore-Sigma | Catalog #: H13303; CAS #: 111-27-3 |
| 1-octen-3-ol | Millipore-Sigma | Catalog # 68225; CAS #: 3391-86-4 |
| Ammonium hydroxide | Millipore-Sigma | Catalog #: 221228; CAS #: 1336-21-6 |
| Benzaldehyde | Millipore-Sigma | Catalog #: 8.01756; CAS #: 100-52-7 |
| Geranyl acetate | Millipore-Sigma | Catalog #: 173495; CAS #: 105-87-3 |
| **Synthetic peptide** | | |
| Synthetic MIP (MIP; EPTWNNLKGMW-amide) | This paper. | N/A |
| **Drosophila parental strains** | | |
| w[*];; GAD1[MI09277] Trojan LexA::QFAD/TM6B, Tb[1] | Bloomington Stock Center | RRID: BDSC_60324 |
| w[*];; ChAT[MI04508] Trojan LexA::QFAD/TM6B, Tb[1] | Bloomington Stock Center | RRID: BDSC_60319 |
| w[*]; VGlut[MI04979] Trojan LexA::QFAD/CyO, PDfd-GMR-nvYFP | Bloomington Stock Center | RRID: BDSC_60314 |
| y[1],w[*], 10xUAS-IVS-mCD8::RFP, 13xLexAop-mCD8::GFP | Bloomington Stock Center | RRID: BDSC_32229 |
| w[*];; GMR32F10-GAL4 | Bloomington Stock Center | RRID: BDSC_49725 |
| hs-FlpG5.PEST;; UAS-MCFO-1 | Bloomington Stock Center | RRID: BDSC_64085 |
| w[*]; 10xUAS-IVS-myr::tdTomato | Bloomington Stock Center | RRID: BDSC_32222 |
| w[*];; 26xLexAop-mCD8::GFP | Bloomington Stock Center | RRID: BDSC_32207 |
| w[*];; 10xUAS-IVS-mCD8::GFP | Bloomington Stock Center | RRID: BDSC_32185 |
| w[-]; 20xUAS-IVS-GCaMP6f | Bloomington Stock Center | RRID: BDSC_42747 |
| w[*]; UAS-SPR-RNAi | Bloomington Stock Center | RRID: BDSC_6888 |
| w[-]; UAS-DenMark, UAS-syt.eGFP; In(3L)D, mirrSaiD[1],D[1]/TM6C, Sb[1] | Bloomington Stock Center | RRID: BDSC_33064 |
| w[*];; 13xLexAop-CD4-tdTomato/ TM6B, Tb[1] | Bloomington Stock Center | RRID: BDSC_77139 |
| y[1],w[*], SPR[MI13553] | Bloomington Stock Center | RRID: BDSC_60934 |
| y[1],w[-]; wg[Sp1]/CyO; 13xLexAop2-6xmCherryHA | Bloomington Stock Center | RRID: BDSC_52271 |
| w[*], dlg[14], frt101/FM7a;; CG11583[c01124], frt80B/TM3, Sb[1] | Bloomington Stock Center | RRID: BDSC_36283 |
| y[1], w[-]; UAS-ANF-GFP | Bloomington Stock Center | RRID: BDSC_7001 |
| w[*]; UAS-DTI | Bloomington Stock Center | RRID: BDSC_25039 |
| SPR[MI13885]-T2A-LexA::QFAD | This paper. | N/A |
| y[1], w[-], SPR-T2A-GAL4 | Shu Kondo, Tohoku University | 83 |
| pebbled-GAL4 (peb-GAL4) | Rachel Wilson, Harvard University | N/A |
| SPR-GAL4::VP16 | Michael Texada, University of Copenhagen | 90 |

**Table 1 (continued) | Sources and identifiers for all key reagents and resources used in this present study**

| Reagent/Resource | Source | Identifier |
|---|---|---|
| **Recombinant DNA** | | |
| pBS-KS-attB2-SA(2)-T2A-LexA::QFAD-Hsp70 | 41 | Addgene Catalog #62949 |
| **Software and algorithms** | | |
| VAA3D (v.3.20) | 125 | RRID: SCR_002609 |
| FluoRender (v.2.26.3) | 127 | RRID: SCR_014303 |
| FIJI (v.2.0.0) | Open-Source | RRID:SCR_002285 |
| R (v.4.1.3) | Open-Source | www.R-project.org |
| R Studio (v.2022.07.2) | Open-Source | www.rstudio.com |
| MATLAB 2021a | MathWorks | www.mathworks.com |
| Python 3 | Open-Source | RRID: SCR_008394 |
| CorelDRAW 2021 | Corel Corp. | www.corel.com |
| Adobe Illustrator 2023 | Adobe Inc. | www.adobe.com |
| SCope | 93 | www.scope.aertslab.org |
| natverse | 28, 142 | www.natverse.org |
| Connectome-neuprint/neuprint-python | Stuart Berg (JRC) | N/A |
| CloudVolume | William Silversmith (Princeton) | www.github.com/seung-lab/cloud-volume |
| ScanImage (v.5.5) | Vidrio Technologies | N/A |

Bruchpilot (Table 1). Hierarchical clustering and principal components analysis (PCA) of glomerular innervation data were performed as previously described[32]. PCA was performed without any arbitrary threshold of significance. Glomeruli and individual MIPergic LN clones were hierarchically clustered using Ward's method ("ward.D2") and Euclidean distance using the "Heatmap" function in the *ComplexHeatmap* package[128]. Pairwise Pearson's correlation coefficient for all possible binary combinations of glomeruli were determined from our MIPergic LN glomerular innervation clonal analysis data using the "cor" function in the base-R *stats* package. These Pearson's correlation coefficients were subsequently assessed for statistical significance by using the "rcorr" function in the *Hmisc* package, which computes a matrix of Pearson's r rank correlation coefficients for all possible pairwise combinations within a data matrix. The *p*-values in this instance are the probability that we would have found a given result if the correlation coefficient was zero (the null hypothesis). This is an indication of whether the aforementioned co-innervation of a given pair of glomeruli is significant or not. In other words, if glomerulus A and B are likely co-innervated by a given MIPergic LN (i.e., positive correlation of MIPergic LN between glomerulus A and B), is this likelihood statistically probable? Further details regarding how significant correlations are computed using this approach are provided in the package's documentation (https://github.com/harrelfe/Hmisc/blob/master/R/rcorr.s). The *corrplot* package was used to create the hierarchically clustered (using Ward's method) representation of these pairwise correlation coefficients (Fig. 1). In every case used, glomerular "odor scene" information is derived from previous assignments[129].

To determine if MIPergic LNs preferentially innervate glomeruli based on valence, glomeruli were assigned "attractive" or "aversive" based on similar assignments previously described[28,129]. These glomerular valences aggregate findings from previous reports[25,130–134], as well as behavioral valence of the odors[135] that glomerulus' OSNs respond to according to DoOR 2.0[136]. Glomeruli whose valence is state-dependent (e.g., the V glomerulus)[137] and DC4 were not included in this analysis. Similar methods were used to determine if MIPergic LNs preferentially innervate glomeruli based on the functional group of a given OR's cognate odorant, with the exception of the V and VM6 glomeruli.

## MIPergic LN ablation experiments

To determine whether MIPergic LNs are the sole source of MIP immunoreactivity within the AL, we used R32F10-GAL4 to drive the expression of a temperature-sensitive variant of Diphtheria toxin (UAS-DTI)[138] in all MIPergic LNs. Animals carrying both transgenes were raised at a permissive temperature of 18 °C, until ~2 days post-eclosion when they were shifted to the non-permissive temperature of ~25–28 °C for ~3 days. After ~3 days at 25 °C or 28 °C animals were processed for MIP-immunolabeling as described above.

## MIPergic LN anatomical marker density analyses

Analysis of syt.eGFP, DenMark, anti-MIP immunoreactive puncta signal, and LN innervation density in antennal lobe glomeruli (via mCD8::GFP signal) was performed as previously described[75]. Images of all antennal lobes within a given brain were collected with similar confocal scan settings (laser power, detector offset, etc.) and later imported into FIJI for quantification. Using the Segmentation Editor plugin and a previously described script (graciously provided by Rachel Wilson, Harvard)[75], ROIs were manually traced every 2–3 slices around the neuropil boundaries of each glomerulus using the anti-DN-Cadherin or anti-Bruchpilot channel, and then interpolated through the stack to obtain boundaries in adjacent slices. To ensure each brain contributed equally when pooling data across brains, signal density values for all glomeruli were normalized to the maximum density value within the given indicator being analyzed (e.g., all density values for syt.eGFP were normalized to the maximum syt.eGFP value). Signal density values were similarly normalized within-indicator, but also within-sex, for assessing sexual dimorphism in MIPergic LN syt.eGFP or DenMark puncta signal (Supplementary Fig. 7). The "ggscatter" function in the *ggpubr* package was used to determine Pearson's correlation coefficients and *p*-values when assessing correlations between effector/anti-MIP and MIPergic LN mCD8::GFP voxel density across all glomeruli. Adjusted $R^2$ values were calculated using the base-R *stats* package and correspond to how well each data being assessed for the given correlation analysis fit a linear model.

## Putative MIPergic LN connectomic analyses − identifying putative MIPergic LNs

All connectome analyses leveraged the publicly available Janelia FlyEM *Drosophila* hemibrain electron microscopy volume (v.1.2.1; https://neuprint.janelia.org/)[60,61], and recently described analysis suites[28,139]. We used several stringent criteria for determining which neurons are most likely MIPergic LNs, the first of which was the candidate neurons must be AL LNs. We then selected those candidate AL LNs that were

previously determined to most likely belong to the patchy AL LN subtype[28]. Candidates were then filtered for those that receive input from the contralaterally projecting, serotonin-immunoreactive deutocerebral (CSD) neurons as all MIPergic LNs express the 5-HT1A serotonin receptor[122], and form connections with the serotonergic CSD neurons[140]. We then used *natverse*[139] to transform the interconnectivity of each candidate neuron into the FlyCircuit whole brain (FCWB) template brain three-dimensional space[141,142], so we could generate a morphological similarity score between our query neuron and neurons FlyLight project's GMR-GAL4 repository[47] by using the built-in NBLAST package (*nat.nblast*)[142]. We selected for only those candidates that achieved a GMR32F10-GAL4 NBLAST score of >0.80, which is greater than the >0.60 score necessary to consider the query neuron and GMR GAL4 neurons identical twins[142]. Lastly, any remaining candidate MIPergic LNs were filtered for those neurons that are considered the hemibrain's highest level of tracing completeness and confidence ("Traced"). Only neurons that met all of these criteria (~7% of all AL LNs) were considered for further analysis.

Despite these stringent criteria, these procedures resulted in the identification of 5 more neurons than there are MIP-immunoreactive neurons in the *Drosophila* AL (14 ideal candidates vs. ~9 MIP-immunoreactive neurons). This means that 5/14 putative MIP LNs analyzed here are likely not MIP AL LNs. As all known biological characteristics of MIPergic AL LNs were exhausted to identify these 14 ideal candidates, there is no convincingly reasonable method to identify which 5 putative MIP LNs should be excluded. Therefore, we elected to analyze all 14 ideal candidates (as opposed to imposing an arbitrary criteria) to ensure any/all MIPergic AL LN connectomic topological properties would be captured. That said, although ~36% of the putMIP LNs analyzed here are likely not MIPergic AL LNs, we generally find similar results for all 14 ideal candidates (Figs. 3, 4 and Supplementary Fig. 3). Thus, we do not find significant distinguishing characteristics of any 5 of these putative MIP LNs that suggests our reported interpretations are incorrectly attributable to their inclusion.

### Putative MIPergic LN connectomic analyses − putMIP LN meshes, segregation indices, and flow centrality
Most methods for analyzing putMIP LN morphology and connectivity have been described recently[28]. Putative MIPergic LN skeleton meshes (Fig. 3g) were fetched from the hemibrain data repository by accessing the neuPrint Python API using the *neuprint-python* (https://github.com/connectome-neuprint/neuprint-python) and *Cloud-Volume* (https://github.com/seung-lab/cloud-volume) packages. The *hemibrainr* package (https://github.com/flyconnectome/hemibrainr) was used to fetch each putMIP LN's metadata and calculate each neuron's dendrite-axon segregation index and flow centrality[68] using the recommended arguments.

### Putative MIPergic LN connectomic analyses − intraglomerular input:output ratio analysis
Glomerular meshes based on PN dendrites were used for all subsequent analyses (input:output ratio by glomerulus, connectivity demographics, etc.)[28]. To establish a input:output ratio for each glomerulus a given putMIP LN innervates, we extracted the number of input and output connections each putMIP LN has within each glomerulus by subsetting the connectors read in from the neuPrint database via the *neurprintr* "neuprint_read_neurons" function. These connectors were then filtered for their presence inside each glomerulus' mesh XYZ coordinate space, segregated based on connection type (e.g., output), then finally summed. By analyzing the data in this manner, as opposed to simply considering the number of putMIP LN axons/dendrites within a given glomerulus, this analysis likely more closely captures putMIP LN input-vs.-output across the AL as *Drosophila* synapses are generally polyadic (reviewed in[62]). To establish a given putMIP LN's input:output ratio across all glomeruli, we used the

following formula: (# of input connections - # of output connections)/ (# of input connections + # of output connections). Therefore, values from −1 to 0 indicate the given putMIP LN sends more output within the given glomerulus. Conversely, values from 0 to 1 indicates the given putMIP LN receives more input within the glomerulus.

### Putative MIPergic LN connectomic analyses − general upstream and downstream demographics analyses
To identify and compare the demographics of each putMIP LN's upstream and downstream partners, putMIP LN connectivity data were first extracted using the *hemibrainr* "simple connectivity" function. The demographic of each presynaptic and postsynaptic partner was generally assigned according to the neuron's accompanying name or type as listed on neuPrint, or by previously established cell type assignments[28]. In cases where a neuron's name or type was unannotated (i.e., NA), the neuron would be categorized as Unknown. The percentage of overall input a given putMIP LN receives from a given neuron category is the sum of connections from a given neuron category to the given putMIP LN divided by the total amount of input that given putMIP LN receives from all categories multiplied by 100%. Similar methods were applied for determining the percentage of overall output a given neuron category receives from a given putMIP LN.

### Putative MIPergic LN connectomic analyses − putMIP LN input polarity analysis
To determine the amount of excitatory, inhibitory, and modulatory input a given putMIP LN receives within each glomerulus, we first categorized each presynaptic neuron as either excitatory, inhibitory, or modulatory based on the presynaptic neuron's neuPrint name/type, previous immunohistochemistry results[31,32,35,143–146], and/or the category assigned in previous reports[28]. However, we acknowledge several caveats to this analysis, such as: (1) this analysis does not account for co-transmission; (2) several glomeruli are truncated within the hemibrain AL[28]; (3) although we consider all LNs as inhibitory as most are either GABAergic or glutamatergic (combined, these represent ~170/ 200 AL LNs)[31,32,35,111,143,145], there are ~4 tyrosine hydroxylase-immunoreactive (dopaminergic) and ~8–15 cholinergic and/or electrically coupled LNs in the AL[32,33,38,39]; (4) although GABA can also act as an intrinsic modulator in the AL (reviewed by Lizbinski & Dacks[147]), we only count GABAergic LNs as part of the "inhibitory input" category here; and, (5) we consider all ventral LNs analyzed here as being glutamatergic, but there are ~4 dopaminergic (tyrosine hydroxylase-immunoreactive) ventral LNs[32]. Once each presynaptic neuron's chemical identity (excitatory, inhibitory, modulatory, or unknown) was determined, we used several approaches to assign these synapses to particular glomeruli. In the case of uniglomerular PNs (uPNs) and OSNs, we leveraged the single glomerulus innervation of these presynaptic neuron types to assign their synapse onto a given putMIP LN synapse to the presynaptic neuron's home glomerulus. That is to say, OSN-to-putMIP LN and uPN-to-putMIP LN synapses were assigned to a glomerulus by: (1) using the home glomerulus assigned to a given presynaptic in the neuron's neuPrint name/ type, or (2) by the home glomerulus assigned to the neuron in previous reports[28]. For instance, if the presynaptic neuron was a cholinergic PN whose home glomerulus is DA2, and this DA2 PN synapses on a given putMIP LN five times, then those five synapses went to the overall excitatory input the given putMIP LN receives within DA2. Neurons were only excluded from this analysis if the presynaptic neuron's home glomerulus was not previously identified[28]. Once the polarity of the input type was established, we used the same methods as above for determining whether the XYZ coordinates of each putMIP LN's synapse(s) with a given presynaptic partner were located in a given glomerulus. Synapse counts for each putMIP LN partner within the given glomerulus were then summed by type (excitatory, inhibitory, modulatory, or unknown), and the

resulting total was divided by the total number of synapses the given putMIP LN makes within that glomerulus to establish percent excitatory, inhibitory input, or modulatory input.

## SPR$^{MII3885}$-T2A-LexA::QFAD generation

The SPR$^{MII3885}$-T2A-LexA::QFAD fly line was established using previously described injections methods[41]. We also note that we also attempted to create an SPR-T2A-GAL4 using the pC-(lox2-attB2-SA-T2A-Gal4-Hsp70)3 construct (Addgene #62957), but no founders emerged (potentially owing to lethality when these construct elements are inserted in the SPR locus). Briefly, pBS-KS-attB2-SA(2)-T2A-LexA::QFAD-Hsp70 and ΦC31 helper plasmid DNA were co-injected into y$^1$, w$^*$, Mi{MIC} SPR$^{MII3885}$. pBS-KS-attB2-SA(2)-T2A-LexA::QFAD-Hsp70 (Addgene plasmid #62949) and pC-(lox2-attB2-SA-T2A-Gal4-Hsp70)3 (Addgene #62957) were gifts from Benjamin H. White (NIH). SPR$^{MII3885}$-T2A-LexA::QFAD transformants were isolated as depicted in Supplementary Fig. 7.

## Single-cell RNA-sequencing (scRNA-seq) analysis of SPR expression

Single-cell transcriptomic data were accessed and downloaded from the SCope web interface (https://scope.aertslab.org) on 03/04/2022. Projection neuron clusters were re-identified as in each dataset's original report[88–90]. Transcript reads were exported log-transformed ($\log(1 + x)$) and reads were counts-per-million (CPM) normalized. Projection neuron subpopulations were then identified within each scRNA-seq dataset using previously established marker genes[88,148,149].

## in vivo calcium imaging − animal preparation

All calcium imaging experiments were performed on female flies ~1–5 days post-eclosion, and at room temperature. All physiology occurred within the animal's ZT0 and ZT8. Animals of the proper genotype were collected and briefly anesthetized on ice. Once anesthetized, an animal was affixed to a custom-built holder with UV curable glue (BONDIC, M/N: SK8024). Our custom-built holder consists of a sheet of aluminum foil with a ~1 × 1 mm square (the imaging window) affixed to a 3D-printed design derived from similar designs described previously[150]. Once mounted, a small window exposing the dorsal side of the brain was created, and covered with twice-filtered recording saline (in mM: 2 CaCl$_2$, 5 KCl, 5 HEPES, 8.2 MgCl$_2$, 108 NaCl, 4 NaHCO$_3$, 1 NaH$_2$PO$_4$, 10 sucrose, and 5 trehalose; adjusted pH: ~7.4)[29]. After establishing the imaging window, the air sacs, fat bodies, and trachea covering the dorsal side of the brain - as well as Muscle 16 - were removed with fine forceps. With the exception of minimal epochs during the synthetic MIP bath application experiments (see below), the brain was continuously perfused with oxygenated (95%O$_2$/5%CO$_2$) recording saline using a Cole-Parmer Masterflex C/L (M/N: 77120-62) at a rate of ~2 mL min$^{-1}$.

## in vivo calcium imaging − image acquisition

For one-photon imaging data (the majority of in vivo physiology data), data were acquired using a Prior Scientific Open Stand (M/N: H175) microscope mounted on Prior Scientific motorized translational stage (M/N: HZPKT1), and equipped with an Olympus 10x/0.30 UPlanFL N objective and an Olympus 60x/1.00 LUMPlanFL N water-immersion objective. A 470 nm CoolLED pE-100 (CoolLED Ltd., Hampshire, UK) was used as the light source. Each trial was captured with a Hamamatsu ORCA-4.0LT camera (Hamamatsu Phototonics, Hamamatsu, Japan), and consists of 40 1,024 × 1,024 frames acquired at a frame rate of ~9 Hz.

A portion of the R32F10-GAL4 odor panel experiments were also acquired using a custom-built two-photon system (Scientifica) equipped with a Mai Tai HP Ti:Sapphire laser (Spectra-Physics) and operated using ScanImage acquisition software (v.5.5; Vidrio Technologies). Emitted fluorescence was captured by a gallium arsenide phosphide (GaAsP) photomultiplier-tube detectors. Each trial consisted of 80 512 × 512 frames acquired at a frame rate of ~3.4 Hz. After data acquisition, a high-resolution z-stack (1,024 × 1,024) was acquired at ~0.21 Hz to enable post-hoc glomerulus identification as previously described[29,40,151–153] (also, see below).

## in vivo calcium imaging − odor preparation and delivery

All odor concentrations are reported as v/v dilutions in paraffin oil (J.T. Baker, VWR #JTS894), or autoclaved and twice-filtered distilled water (for diluting acids). For example, $10^{-2}$ dilution indicates that one volume of an odor is diluted with 100 volumes of paraffin oil.

For one-photon imaging data (the majority of in vivo physiology data), dilutions were prepared in 2 mL odor vials (SUPELCO; P/N: 6020) that contained a final volume of 1 mL of diluted odor in paraffin oil every other day, or after two experiments (whichever came first). Odors were generally presented as previously described[75,77,119]. Briefly, a carrier stream of carbon-filtered, dehumidified, air was presented at 2.2 L min$^{-1}$ to the fly continuously through an 8 mm Teflon tube placed ~1 cm away from the fly. A three-way solenoid (The Lee Company, P/N: LHDA1231315H) diverted a small portion of the airstream (0.2 L min$^{-1}$) through the headspace of an odor vial for 200 ms after triggering an external voltage command (TTL pulse) at frame 20 of the trial. Considering the above, the odor is diluted further (by 10-fold) prior to delivery to the animal. The odor stream joined the carrier stream 11 cm from the end of the tube, and the tube opening measured ~4 mm. Odor delivery during two-photon imaging was similar, but differed slightly in that: (1) odor cartridges (see below) instead of a 2 mL odor vial; (2) the continuous airstream was presented via a custom-built glass tube; and, (3) the TTL pulse occurred at frame 30 of the trial.

Methods for assessing preparation health and performing multiple odor trials conform to previous work[75,119]. At the start of each experiment, the animal was presented a test odor ($10^{-3}$ 2-heptanone) to assess the preparation's health. Only the data collected from animals whose responses to this test odor were robust and did not dramatically change from baseline over the course of the experiment were used for further analysis. The only exceptions to this were those data collected in synthetic MIP bath application experiments (see below), since bath application of any modulator would likely result in network property changes that would consequently change olfactory responses. Therefore, the test odor was only initially presented to those animals used for synthetic peptide application experiments, so their initial olfactory response health could be assessed. Each experiment consisted of multiple odor trials (3 for OSNs; 4 for LNs) within a preparation which were then averaged to attain a within-animal response. These within-animal averages were subsequently averaged across many animals for subsequent statistical analysis, and "$n$" is reported as the number of animals. Each odor trial consisted of five 200 ms pulses of odor with a 1 ms interpulse interval. The same odor was never presented twice within 2 min to prevent depletion of the odor vial's headspace. If multiple odors were to be tested, then they were presented randomly. If multiple concentrations of a given odor were to be tested, then the lower concentration was presented before the higher concentration. Air entered and exited each odor vial through a PEEK one-way check valve (The Lee Company, P/N: TKLA3201112H) connected to the vial by Teflon tubing. The odor delivery tube was flushed with clean air for 2 min when changing between odors/concentrations. As an additional preemptive measure, all odor delivery system components were hooked up to the house vacuum line overnight. The olfactometer used in two-photon data collection consisted of odor cartridges (a syringe housing a piece of filter paper that was doused in 10 μl of diluted odor) hooked into a custom glass carrier stream delivery tube as previously described[154].

## in vivo calcium imaging − data analysis

All calcium imaging data were analyzed using a custom-made MATLAB script graciously provided by Marco Gallio (Northwestern University)

and has been described previously[51,155,156]. With the exception of any preparations that violated the aforementioned criteria (e.g., movement, diminishing prep health, etc.), no data points or outliers were excluded from our analyses. Generally, the number of flies to be used for experiments are not a limiting factor, therefore no statistical power analyses were used to pre-determine sample sizes. Regardless, our sample sizes are similar to those in previous reports that perform similar experiments[30,51,157–162]. Before analyzing the data, a Gaussian low-pass filter (sigma=1), bleach correction (exponential fit), and image stabilizer algorithms were applied to the given trial's raw ΔF/F signal. Similar preprocessing for two-photon microscopy data was similar, with the exception of a higher sigma during Gaussian low-pass filtering (sigma=2). A trial's average fluorescence image was used as a guide to draw consistently sized circular regions-of-interest (ROI) within a given glomerulus. Calcium transients (ΔF/F) within the ROI were measured as changes in fluorescence (ΔF) normalized to baseline fluorescence (F, fluorescence intensity averaged across 2 sec just prior to odor onset). Within-animal responses were established by averaging across several odor trials in the given preparation (3 for OSNs; 4 for LNs). These within-animal responses were then pooled for each stimulus identity and concentration across animals. These pooled averages were used for all subsequent statistical analyses and the "n" is reported as the number of animals. Glomeruli were manually identified post-hoc by comparing acquired images to well-defined three-dimensional maps of the AL[163,164]. Only the glomeruli that were reasonably identifiable were considered for analysis.

## Myoinhibitory peptide (MIP) application experiments

MIP (MIP; EPTWNNLKGMW-amide) was custom made by GenScript (Piscataway, NJ, USA) at the highest purity available (>75%). The sequence we chose to use for MIP is identical to the sequence previous investigations have used when discerning the role of MIP in the *Drosophila* circadian system[158]. In pilot experiments, we tested another sequence of MIP (RQAQGWNKFRGAW-amide) that was previously detected at the highest abundance by direct profiling of single ALs using mass spectrometry[26,165]. Experimental results produced using peptide of either sequence were not qualitatively different, but all results reported here use the MIP previously used in circadian studies[158]. To test how MIP application adjusts odor-evoked responses, a 1,000 μM working solution was made by diluting a small portion of the lyophilized peptide in nuclease-free water (ThermoScientific, #R0581). After testing the initial odor-evoked responses of the neurons being tested for a given experiment, the perfusion system was momentarily switched off so a small portion of our MIP working solution could be pressure injected into the AL to a final concentration of 10 μM. This final concentration was chosen for several reasons, which include: (1) we wished to remain consistent with other studies of peptidergic modulation in the *Drosophila* AL[29,30,157]; (2) we wished to be consistent with studies on the effects of MIP in other circuits[158]; and, (3) previous reports have already determined that our chosen effective concentration (10 μM) is the optimal concentration for testing the effect of MIP on *Drosophila* neurons[158]. Ten minutes after MIP pressure injection, the animal's odor-evoked responses were tested as before MIP injection, and then the perfusion system was switched back on. Ten minutes after turning the perfusion system back on, the animal's odor-evoked responses were once again tested as they were initially. Re-testing the animal's response to the test odor (10⁻³ 2-heptanone) at the end of these experiments could not be used as a reliable means for assessing prep health due to changes in circuit member responses induced by modulator bath application. Therefore, for these experiments no animal was tested for longer than the average time that animals were reliably healthy in the MIPergic LN odor panel experiments (~90 min). Furthermore, we believe these preparations remain healthy throughout the entire experimental epoch as ACV responses increase or do not significantly diminish over the course of the

experimental epoch in many glomeruli (Fig. 1). It is notable that subtle differences exist between the"pre-MIP"/baseline peb-GAL4 GCaMP responses (Fig. 1) and peb-GAL4 > SPR-RNAi GCaMP responses (Fig. 8). The difference in baseline across these figures likely stems from: (1) subtle differences in the imaging plane which is inherent to these types of in vivo recordings across these experiments; and, (2) the difference in genotype of the animals being recorded in each respective figure. To address this, the within-figure baseline values are only ever internally compared by having within-animal measures across treatments (i.e., pre-MIP, MIP, and post-MIP). In doing so, the impact of MIP is with relation to the values from the same animal.

## Quantification and statistical analyses

**General approach.** Statistical analyses were performed using R (v.4.1.3) in R Studio (v.2022.07.2). Values to be analyzed were concatenated in Excel before importing into the relevant analysis software. Statistical results are reported within the main text and/or figure legends. All statistical tests were two-tailed. All boxplots display the minimum, 25th-percentile, median, 75th-percentile, and maximum of the given data. Additional analysis details are provided for each set of experiments above. Where possible, values are given as mean ± SEM. Statistical significance is defined as: *$p \leq 0.05$, **$p \leq 0.01$, ***$p \leq 0.001$.

**Statistical analyses related to neuroanatomical experiments.** The *ComplexHeatmap* package was used to hierarchically cluster glomeruli and individual MIPergic LN clones using Ward's criteria and Euclidian distance. The *ClustVis* package (https://github.com/taunometsalu/ClustVis)[117] was used to perform PCA on individual MIPergic LN innervation patterns. The "cor" function in the base-R *stats* package and the "rcorr" function in the *Hmisc* package were used to calculate statistically significant Pearson's correlation coefficients for MIPergic LN pairwise glomerular innervation patterns. The *ggpubr* package's "ggscatter" function was used to determine Pearson's correlation coefficients and *p*-values when assessing correlations between: (1) effector/anti-MIP and MIPergic LN mCD8::GFP voxel density across all glomeruli, and (2) MIPergic LN glomerular innervation frequency as a function of each glomerulus' volume. Adjusted $R^2$ values were calculated using the base-R *stats* package and correspond to how well each data being assessed for the given correlation analysis fit a linear model. The Shapiro-Wilk test (the *rstatix* package's "shapiro_test" function) was used to evaluate any deviations from a normal distribution. Welch's unpaired *t*-test was used to determine if MIPergic LNs preferentially innervate glomeruli based on inferred hedonic valence. A Kruskal-Wallis rank sum test followed by pairwise Bonferroni's-corrected Dunn's multiple comparisons test was used to determine if: (1) MIPergic LNs preferentially innervate based on the functional group found along the odorant that activates the given glomerulus' odorant receptor; (2) SPR-GAL4::VP16 expression in antennae and maxillary palps significantly differs between males, mated females, and virgin females; (3) SPR-GAL4::VP16 expression in glutamatergic LNs between males, mated females, and virgin females; (4) SPR-T2A-GAL4 expression in maxillary palps significantly differs between males, mated females, and virgin females; and, (5) the number of MIPergic LNs differ between males, mated females, and virgin females. Welch's one-way ANOVA with a Bonferroni multiple comparisons correction was used to assess statistically significant differences in SPR-T2A-GAL4 expression in antennae between males, mated females, and virgin females. A two-way ANOVA with a Greenhouse-Geisser sphericity correction followed by a Bonferroni's multiple comparisons test was used to assess sexual dimorphism in MIPergic LN syt.eGFP or DenMark puncta density across glomeruli.

**Statistical analyses related to physiology experiments.** Background-subtracted changes in fluorescence over time (ΔF/F) analyses were carried out using custom MATLAB scripts previously

described[51,155], and are represented as individual traces overlaid by the mean with dilutant-only (e.g., paraffin oil-only) responses subtracted. Peak response (Fig. 3e) refers to the maximal $\Delta F/F$ value within the time of odor onset to ~1 s post-odor onset averaged across all animals. Area under the $\Delta F/F$ curve (AUC) was modified from previous reports[166], such that AUC was calculated using Simpson's rule ("sintegral" function in the *Bolstad2* package) as the integral of the $\Delta F/F$ traces from the beginning until 1 s after odor delivery with a baseline of 1 s before stimulus onset. To assess OSN odor-evoked response differences across MIP treatments, we first determined if normality could be assumed (as above). If normality could be assumed, then an omnibus repeated measures one-way ANOVA with a Greenhouse-Geisser sphericity correction was performed (RM one-way ANOVA) ("anova_test" function in *rstatix*). If significant differences were detected with the omnibus, then pairwise repeated measures *t*-tests (RM *t*-tests) with a Holm multiple comparisons correction were performed to identify which groups were statistically different. If normality could not be assumed, then a Friedman rank sum test followed by Holm-corrected paired two-sided Wilcoxon signed-rank test was performed. All effect sizes reported were calculated using either the "cohens_d" (for parametric data) or "friedman_effsize" (for non-parametric data) function from the *rstatix* package, which compute Cohen's *d* or Kendall's *W*, respectively.

### Reporting summary

Further information on research design is available in the Nature Portfolio Reporting Summary linked to this article.

## Data availability

Connectomic and scRNA-seq source data are available on neuPrint [https://neuprint.janelia.org/] and SCope [https://scope.aertslab.org/], respectively. All original light microscopy confocal scans used to create the figure panels are available at https://zenodo.org/10.5281/zenodo.8127341. Source data are provided with this paper.

## Code availability

With the exception of code that was graciously provided to us by others, all code that was used to analyze or plot data is available at https://github.com/tsizemo2/Sizemoreetal2023.

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

## Acknowledgements

We are grateful to Kristyn Lizbinski, Eric Horstick, and John Carlson for the helpful comments given on earlier manuscript drafts. Flies were kindly shared or acquired from: Shu Kondo, David Krantz, Michael Texada, Jim Truman, Rachel Wilson, Quentin Gaudry, the Bloomington *Drosophila* Stock centers (NIH P40OD018537), and the Janelia Fly Light project. Christian Wegener kindly provided the MIP antibody that was developed by Manfred Eckert. Marco Gallio and Rachel Wilson graciously provided analysis scripts. Karen Menuz gave helpful insight in the interpretation of RNA-sequencing results from prior work. Stephen Plaza, Alexander Bates, Marta Costa, Greg Jefferis, and Philipp Schlegel gave helpful advice and instruction regarding connectomic analyses, and graciously shared odor scene and valence information. Our connectomic analyses also benefited from discussions in the nat-user community, and a workshop organized by the Virtual Fly Brain and *Drosophila* connectomics teams (virtualflybrain.org; WT 208379/Z/17/Z). Fengqiu Diao gave technical advice for creating the SPR$^{MI13885}$-T2A-LexA::QFAD Trojan exon transgenic animals. Kristyn Lizbinski and James Jeanne graciously provided advice and equipment specifications that lead to the construction and use of the olfactometer system described here. Kristyn Lizbinski, James Jeanne, Marco Gallio, Mehmet Keles, and Gaby Maimon provided invaluable technical advice for performing in vivo physiology. Kevin Daly kindly loaned equipment to us for our purposes and gave invaluable technical advice regarding their use. We would also like to thank Jacob Ralston for assisting by collecting confocal scans of AL anti-MIP labeling from R32F10-GAL4 diphtheria toxin ablation experiments. This work was supported by a Grant-In-Aid of Research (G20141015669888) from Sigma Xi, The Scientific Research Society (T.R.S.), start-up funds from WVU (A.M.D.), a National Institutes of Health R03 DC013997 (A.M.D.), a National Institutes of Health R01 DC016293 (A.M.D.), and two AFOSR DURIP awards (FA9550-19-1-0179 and FA9550-20-1-0098) (A.M.D.).

## Author contributions

T.R.S. conceived and implemented all experiments, analyzed the subsequent data, prepared figures, and wrote and edited the manuscript. J.J. acquired data that contributed to Fig. 5. A.M.D. conceived experiments, acquired funding, and helped with manuscript edits.

## Competing interests

The authors declare no competing interests.
