## [Peer Review File · Nature Communications]

Heterogeneous Receptor Expression Underlies Non-uniform Peptidergic Modulation of Olfaction in *Drosophila*REVIEWER COMMENTS

Reviewer #1 (Remarks to the Author):

Neuropeptidergic modulation of sensory systems is responsible for the vast range of potential responses to individual stimuli depending on internal state, environmental condition, and a host of other factors. But how a single neuropeptide could mediate multiple different kinds of responses, even to the same odorant, is unknown. Sizemore and colleagues use the powerful *Drosophila* antennal lobe system to assess how MIP, the myoinhibitory peptide, mediates differential adjustments to input based on olfactory channel. This manuscript is a considerably data-heavy manuscript, which is commendable, but despite the considerable data, it does not live up to the stated claims. The work is an important anatomical exploration of MIP-positive local interneurons, as the authors identify that the likely source of MIP is from patchy GABAergic LNs, and thoroughly characterize the input and output relationships of those LNs using recent hemibrain connectomics assessments. Further, there are important characterizations of MIP-response within different glomeruli. Where the paper falls short, however, is in proving its assertions about MIP and function through the sex-peptide receptor (SPR). Many of the conclusions feel overstated; the top-notch anatomical and ultrastructural characterization is accompanied by comparatively weaker functional analyses that are suggestive of the conclusions stated by the authors, but leave essential alternative hypotheses unexplored, leading to overstatements in conclusion.

This is an important anatomical work that is absolutely worth publication, but considerable additional experiments or revision and restatement of the conclusions would be essential before being acceptable for publication in *Nature Communications*.

Specific Comments:

1) The analysis of an incredibly important tool, the SPR-T2A-GAL4 driver (and with it, the SPR-positive neurons), feels very preliminary compared to the thorough assessment of the MIP-positive neurons. It would benefit from additional analysis, better images (especially in the case of panels like 7g), and a deeper discussion of the role that SPR-positive glia may have (or reduced discussion – right now, the glia are a distraction and it is unclear why the authors discuss them so little in the manuscript but lend a considerable amount of figure panels to the cells).

2) The authors have a challenge in proving that MIP signaling through SPR in the ORNs is the culprit of neuropeptidergic regulation. Figure 8 is suggestive of a role, but the authors do not assess other potential receptors, how co-release with GABA might influence and narrow down the effects, and indeed, whether it is even release by the patchy LNs that underlie this phenomenon. There are a few potential experiments that might get at this including GAD1 RNAi in the patchy LNs to see if GABA plays a role in the response to MIP (examining co-release), or blocking release in general from patchy LNs via shibire or tetanus toxin. The model expressed in Figure 8e leaves too much to be desired and many critical avenues unexplored.

3) I cannot understate how important and thorough the anatomical and connectomic characterization of the MIP-positive neurons is to the field. This is the true strength of the paper. The functional experiments are interesting, but raise a lot more questions than they answer. The authors may wish to consider reframing the paper to focus on the anatomy and then using the functional data to extend potential points, where necessary. This may aid in the presentation and the logic, as with the two elements, the paper feels a bit frenetic in its logic and like two stories inelegantly combined.

4) The values between Figure 1 and Figure 8, which are directly examinations of MIP application to the antennal lobe and then optical recordings from ORNs of different glomeruli, are quite different. Why are the values for so many of these seemingly analogous experiments so vastly different? This raises severe questions about the validity of the experiments and the absence of effect in the SPR RNAi experiments (i.e., if the same control data is not observed, is there something wrong with the experiment itself).

Reviewer #2 (Remarks to the Author):

The goal of Sizemore et al is to understand how a single peptide that appears to be uniformly distributed across all olfactory centers can have stimulus-specific effects on sensory stimuli. To do this they investigate the cellular, physiological and structural substrates that enable myoinhibitory peptide (MIP) to differentially modulate olfactory input to distinct olfactory glomeruli in the *Drosophila* brain.

Sizemore et al provide thorough and rigorous characterization of MIP-expressing neurons and their targets in the antennal lobe. While it doesn't directly explain how MIP is necessary and sufficient to simulate the drive of *Drosophila* toward food odors with the context of satiation, or how it influences odor preferences post-mating, it provides functional electrophysiology and imaging data describing how MIP influences olfactory glomeruli activity. Very few studies provide the functional data that explains how a peptide influences multiple olfactory glomeruli and this manuscript provides an important framework for understanding how neuropeptides can influence ethologically-relevant behaviors. This manuscript is an important addition to the neuromodulatory literature as it can be used as a model for not only how other peptides influence sensory behavior in *Drosophila*, but can also be extended to other, more complex brains since so little is understood about how peptides innervate and affect the function of discrete brain regions.

I have very few comments as I found the manuscript thorough, understandable and the language clear and concise. The methods are thorough and sufficient to replicate the work. The supplemental data is extensive and adds context and rigorous analysis to the main body figures and text. I also applaud the authors on their rigorous representation of raw data in all of the figures, use of color-blind friendly images, and for providing the crossing schemes for the more complicated genetics performed.

Major comments:

The only data the authors do not provide to support their hypothesis is a demonstration of the direct role of endogenous MIP in the MIP neurons. This would be possible by activating the MIP neurons (via olfactory stimulus or artificially if feasible) when MIP levels are reduced (for example by RNAi) then demonstrating this affects the functional role of MIP on post-synaptic neurons. However, the authors state that they tried to do a similar experiment, but were unable to detect any change in the postsynaptic neurons. Since the authors show rigorous and convincing data demonstrating how MIP influences activity of antennal glomeruli, and a functional role for the MIP receptor (sex-peptide receptor), I believe this renders demonstrating the endogenous MIP effects unnecessary and out of the scope of the current manuscript. However, should the authors want to determine the causal and/or coordinated effects of co-transmission of the MIP neurons on behavior or physiology in a future project/manuscript, this might be a helpful experiment.

Minor comments:

- 1) The use of 'olfactory channels' in the intro is confusing – easy to get it confused with olfactory receptors. Recommend rephrasing to be more specific (I think antennal glomeruli will work in this context).
- 2) Please clarify whether there were individual differences in SPR-Gal4 neuron expression similar to that seen with MIP neuron expression, or whether the expression of SPR-expressing neurons was consistent from fly to fly. I understand this might be difficult since the T2A line will show all neurons where SPR gene is expressed and SPR protein might be expressed, whereas your analysis with MIP antibody demonstrates where MIP protein is expressed. Clarifying this might be helpful to the reader.

Reviewer #3 (Remarks to the Author):

In the manuscript by Sizemore and colleagues, the authors perform an impressive series of experiments to characterize the function of the MIP neuropeptide and its receptor SPR in the *Drosophila* antennal lobe (AL). MIP can modulate olfactory behaviors and the authors show it can modulate responses from olfactory neurons. Using co-staining with MIP antibodies and genetic labeling of AL-innervating neurons, the authors identify ~9 GABAergic LNs as the only neurons with the AL circuit to express MIP. By using mosaic labeling of these 9 MIP LNs, they identify them as patchy LNs. They further identify the innervation pattern of all MIP LNs and find they, like other patchy LNs, differ in their innervation patterns from animal to animal but as a whole still cover the entire AL. The authors present a comprehensive analysis of the glomerular input/output of the MIP neurons and do not identify differences that could explain the functional differences observed. They next use their anatomical identification of MIP LNs to query the EM reconstruction of the *Drosophila* brain, and identify 14

putative MIP neurons in this reconstruction. This affords them additional levels of analyses to even more comprehensively characterize the input/output relationships of MIP LNs. Their analyses again support the hypothesis that MIP neuron innervations cannot explain differential responses to MIP, prompting the authors to generate a new knockin of the SPR gene to capture its expression pattern. This reveals SPR heterogeneous expression in OSNs, LNs, and PNs, suggesting differential responses to MIP are due to differential expression of SPR. This is supported by RNAi knockdown of SPR in OSNs, which affected MIP-mediated functional effects observed in some but not all glomeruli.

This is a very impressive body of work aimed at investigating how an apparently 'simple' neuromodulatory system with a single neuropeptide and a single receptor affects the function of olfactory signaling. The data is robust and comprehensive. The methods are well explained and appropriate. The writing is clear and easy to follow. The genetic methods and steps employed to uncover the function of MIP are likely to become a guide for future investigators wishing to study the anatomy and function of other neuromodulators. I am enthusiastic about this study, and have only minor concerns that can be addressed without the need of additional experiments.

1. Figure 1a. Just visually comparing pre vs MIP injection GCAMP6 signals in DM1 and DM4, it appears odor responses go down not up as reported? Is the significant increase in response shown in (b) due to a decrease in basal fluorescence so that the ΔF increases? Or perhaps a more representative image should be shown?

Minor.

1. Please define MIP in the Figure 1 legend for its first appearance in figures (instead of in Figure 2 legend).
2. Page 2, right column. Please define the first use of MIP-ir here (and not on page 4, top left column as it is now).
3. Page 6, bottom of left column/top of right column. "Thus, we first used several criteria (see Methods) to identify fully reconstructed putMIP LNs," Given the importance of these criteria in the interpretation of the identified putMIPs, please summarize these criteria here in the main text. I.e, the criteria included identifying patchy LNs that receive synaptic inputs from CSB neurons, y, z, etc.
4. 9 MIP neurons were identified by genetic labeling, and 14 putative MIP neurons were identified in the EM reconstruction based on criteria mentioned above. Does this mean 5/14 are likely not MIP neurons? Or that additional MIP neurons might be present that were not genetically labelled? Some discussion on the difference in numbers would be helpful.
5. Figure 5. Please mention the genotype used for these experiments in the main text or figure legend.
6. Page 9, left column. Please define the first use of "DCV" here.
7. SPR is described as an 'inhibitory receptor'. Please clarify somewhere in the main text (intro, discussion, etc.) what is known regarding how it functions as an inhibitory receptor. For example, is it a ligand-gate ion channel? Or does it trigger Gi signaling?

Reviewer #1 (Remarks to the Author):

*Neuropeptidergic modulation of sensory systems is responsible for the vast range of potential responses to individual stimuli depending on internal state, environmental condition, and a host of other factors. But how a single neuropeptide could mediate multiple different kinds of responses, even to the same odorant, is unknown. Sizemore and colleagues use the powerful *Drosophila* antennal lobe system to assess how MIP, the myoinhibitory peptide, mediates differential adjustments to input based on olfactory channel. This manuscript is a considerably data-heavy manuscript, which is commendable, but despite the considerable data, it does not live up to the stated claims. The work is an important anatomical exploration of MIP-positive local interneurons, as the authors identify that the likely source of MIP is from patchy GABAergic LNs, and thoroughly characterize the input and output relationships of those LNs using recent hemibrain connectomics assessments. Further, there are important characterizations of MIP-response within different glomeruli. Where the paper falls short, however, is in proving its assertions about MIP and function through the sex-peptide receptor (SPR). Many of the conclusions feel overstated; the top-notch anatomical and ultrastructural characterization is accompanied by comparatively weaker functional analyses that are suggestive of the conclusions stated by the authors, but leave essential alternative hypotheses unexplored, leading to overstatements in conclusion.*

This is an important anatomical work that is absolutely worth publication, but considerable additional experiments or revision and restatement of the conclusions would be essential before being acceptable for publication in Nature Communications.

Specific Comments:

1) The analysis of an incredibly important tool, the SPR-T2A-GAL4 driver (and with it, the SPR-positive neurons), feels very preliminary compared to the thorough assessment of the MIP-positive neurons. It would benefit from additional analysis, better images (especially in the case of panels like 7g), and a deeper discussion of the role that SPR-positive glia may have (or reduced discussion – right now, the glia are a distraction and it is unclear why the authors discuss them so little in the manuscript but lend a considerable amount of figure panels to the cells.

> Our goal with the analysis of the SPR-T2A-GAL4 (& the SPR-GAL4::VP16 driver) was to provide a resource for the entire community of researchers interested in studying SPR signaling in olfaction, while still staying relatively grounded in the main interests of this study (OSN SPR expression). As a result, we focused our efforts on delineating the home-glomerulus for SPR-T2A-GAL4⁺ OSNs, but we also provide a “full-disclosure” of other SPR-T2A-GAL4⁺ in the AL when/where possible.

To address the reviewer’s comment, we have: (1) relocated the SPR-T2A-GAL4⁺ glia panels to **Supplementary Fig. 4**, and (2) included new higher-resolution images for panels 7g, 7h, and 7i. For clarity, we chose to place these panels in the supplementary figures (as opposed to completely removing them) so readers would be

less likely to become distracted by it, but would still have access to this information.

2) *The authors have a challenge in proving that MIP signaling through SPR in the ORNs is the culprit of neuropeptidergic regulation. Figure 8 is suggestive of a role, but the authors do not assess other potential receptors, how co-release with GABA might influence and narrow down the effects, and indeed, whether it is even release by the patchy LNs that underlie this phenomenon. There are a few potential experiments that might get at this including GAD1 RNAi in the patchy LNs to see if GABA plays a role in the response to MIP (examining co-release), or blocking release in general from patchy LNs via shibire or tetanus toxin. The model expressed in Figure 8e leaves too much to be desired and many critical avenues unexplored.*

> The reviewer raises several interesting points, each of which we address below and have clarified within the revised manuscript.

First, in regard to the concern that we do not assess other potential receptors, SPR is the only known receptor for MIP (see Hansen et al 2011, *J Biochemical and Biophysical Research Commun*; Kim et al., 2010 *PNAS*). Although it is possible that MIP may activate another GPCR, there is no published evidence along these lines of which our group is aware.

Second, regarding the concern that we do not assess GABA & MIP co-transmission, we agree that understanding the contribution of MIPergic LN-derived MIP vs. GABA is an important topic. While we greatly appreciate the reviewer's experimental suggestions, each would require the ability to mobilize MIP release from MIPergic LNs via artificial activation. However, as was stated in our initial submission and is now revised (see below), it is not currently technically feasible to determine the amount of artificial activation necessary to mobilize MIP release from MIPergic LNs. That is to say, it is not currently possible to determine whether we are actually triggering the release of MIP from MIPergic LNs using artificial activation methods. Therefore, any observation acquired from knocking down GAD1 in MIPergic LNs while recording OSN GCaMP signals before and after artificially activating MIPergic LNs cannot be directly attributable to MIPergic LN-derived MIP.

We agree with the reviewer that MIPergic LN-mediated GABA and MIP co-transmission is an important remaining avenue from this research. However, to adequately investigate this would represent an experimental undertaking that typically warrants a manuscript of its own (e.g., Sherer et al., 2020, *PLOS Genetics*). Moreover, the results from such experiments would be tangential to the goal of our study (to understand the non-uniform nature of the effects of MIP), and therefore we do not feel their inclusion would add to the strength of this current manuscript. For clarity, the effects of MIP physiologically delineated here are strictly observed after exogenous MIP application. That is to say, none of these results arise from recordings in which MIPergic LNs are artificially activated while recording from OSNs. Thus, the most parsimonious interpretation of our results is that our observations arise from the actions of the exogenous MIP applied, as opposed to endogenous MIPergic LN release of GABA that is then acting to influence the effects of the exogenous MIP. Furthermore, we no longer observe the effects of MIP on WT OSNs if we knockdown SPR in OSNs. As this manipulation only changes MIP's ability to induce neuronal changes, and leaves

GABA signaling untouched, the most parsimonious interpretation is that MIPergic LN-derived GABA does not influence the effects of MIP activation of MIP-SPR signaling along OSNs.

As stated above, we attempted to determine the level of artificial activation necessary to cause MIP release from MIPergic LNs during the course of our investigation. These attempts were described in our initial submission under the subsection entitled “Functional implications of GABA and MIP co-transmission from MIPergic AL LNs” in the Discussion section. We acknowledge this section could be more clearly conveyed, and therefore we have rewritten this subsection (lines 385-424) so it reads:

“We have shown that a small ensemble (~5% of all AL LNs) of GABAergic AL LNs are the sole source of MIP in the *Drosophila* AL (**Fig. 2 and Supplementary Fig. 1**). This implies that MIPergic LNs have the capacity to adjust AL olfactory processing through both GABA- and MIP-release. Previous works found that the iono- and metabotropic GABA receptors are expressed amongst all AL principal neuron types^{27,29,108}, and we show SPR is similarly expressed by members of every AL principal neuron type (**Fig. 7**). Therefore, MIPergic LN activation could plausibly cause both fast-acting and long-lasting inhibition in the same and/or disparate downstream target. Moreover, MIPergic LN-derived GABA and MIP may simultaneously act on the same downstream target(s) to synergistically modulate their activity to have a greater effect than either modulator alone might achieve. Alternatively, MIPergic LNs might primarily use GABA throughout the course of ongoing network activity, and use MIP only under special circumstances (see example below).

We attempted to parse the contribution of GABA and MIP over the course of our investigation. We conceived an experimental paradigm in which MIPergic LNs would be artificially activated chemogenetically via P2X2 misexpression¹⁰⁹ and ATP injection, while we recorded OSN GCaMP responses before and after MIPergic LN activation. The results of these experiments would then be compared to similar experiments wherein MIPergic LN GAD1 expression would be knocked down as a means of resolving the contribution of MIPergic LN-derived GABA. However, both approaches require prior knowledge of what level of artificial activation is necessary to mobilize MIPergic LN dense core vesicles (DCVs), and thus MIPergic LN-derived MIP release.

To resolve the minimal strength of artificial activation necessary to mobilize MIPergic LN DCVs, we artificially activated MIPergic LNs (described above), while simultaneously recording DCV changes via either ANF-GFP⁴⁸ or the neuropeptide release sensor NPPR-ANP-GCaMP6s¹¹⁰. However, we were unable to detect any change in either indicator even when we injected 100mM ATP - a concentration 10x-greater than what is necessary to activate other AL LNs¹¹¹. As a consequence, it remains infeasible to simply artificially activate MIPergic LNs, while measuring a downstream neurons' responses, and accurately attribute changes in the downstream neuron MIP released from MIPergic LNs.

Co-transmitters often increase the computational capacity of the neuron they are released from, and the plasticity of the networks in which they act^{112–115}. Therefore, it is likely that future investigations using novel technology may find that the MIPergic LNs play a far greater role in AL processing than what is resolved here. As detailed above, each co-transmitter may confer unique and substantial contributions to the overall role

these interneurons have in AL processing. As such, this aspect of MIPergic LN function is an important and remaining question that should be addressed by future investigations.”

Finally, there are two issues with the supposition that MIP release from another source might underlie our observations. The first issue is that our experiments regarding the effect of MIP on OSNs (**Figs. 1 & 8**) involve exogenous application of MIP (rather than endogenous release). Unlike artificial activation of all/many neurons, this approach – as detailed above - ensures any changes in OSN responses we observe do not arise from other transmitters released by the MIPergic LNs. The second issue is that the only MIP-immunoreactive neurons in the AL are the MIPergic LNs (**Fig. 2**). If these MIPergic AL LNs are ablated, and all other MIPergic neurons are left intact, all MIP-immunoreactivity is lost in the AL – the only brain region in which OSN axons reside (**Fig. 2**). Moreover, there is no MIP-immunoreactivity in the peripheral olfactory appendages where OSN cell bodies and dendrites reside (**Supplementary Fig. 1**). Taken together, the most parsimonious interpretation of these results are that MIPergic LNs are the sole endogenous source of MIP within the AL.

Taking all of the above together, we have done as the reviewer requested and revised the language used when discussing our results to explicitly distinguish the effects of the relevant patchy AL LNs and the effects of MIP as a singular signaling molecule.

3) I cannot understate how important and thorough the anatomical and connectomic characterization of the MIP-positive neurons is to the field. This is the true strength of the paper. The functional experiments are interesting, but raise a lot more questions than they answer. The authors may wish to consider reframing the paper to focus on the anatomy and then using the functional data to extend potential points, where necessary. This may aid in the presentation and the logic, as with the two elements, the paper feels a bit frenetic in its logic and like two stories inelegantly combined.

> We thank the reviewer for their compliment, and acknowledge their concern regarding the manner in which we report our functional, anatomical, and connectomic evidence. We have reframed the context in which we report the impact of exogenously applied MIP relative to the nature of the neurons from which endogenous MIP derives. By addressing the specific concerns raised above, we hope that the additional clarifying statements and other edits included in this revised manuscript assuage the reviewer’s critique of our initial submission.

4) The values between Figure 1 and Figure 8, which are directly examinations of MIP application to the antennal lobe and then optical recordings from ORNs of different glomeruli, are quite different. Why are the values for so many of these seemingly analogous experiments so vastly different? This raises severe questions about the validity of the experiments and the absence of effect in the SPR RNAi experiments (i.e., if the same control data is not observed, is there something wrong with the experiment itself).

> The reviewer raises an important point, which we have taken very seriously and hopefully have assured the reviewer of any questions they might have regarding these data below.

The most likely rationale for subtle differences between figures 1 & 8 is inherent to the difference in genotype of the animals being recorded for each respective figure. For clarity, the animals in figure 1 are *peb-GAL4* driving the expression of GCaMP, while those in figure 8 are *peb-GAL4* driving the expression of GCaMP & SPR-RNAi. For this reason, one would not expect to observe identical baseline values (i.e., identical $\Delta F/F$ in the pre-MIP treatment across Figure 1 & 8). To address this, the within-figure baseline values are only ever internally compared by having within animal measures across treatments (i.e., pre-MIP, MIP, post-MIP). In this way, the impact of MIP is with relation to the values from the same animal.

In addition to this, subtle differences in the focal plane inherent to these types of *in vivo* recordings likely contribute to the differences between the two figures.

Reviewer #2 (Remarks to the Author):

The goal of Sizemore et al is to understand how a single peptide that appears to be uniformly distributed across all olfactory centers can have stimulus-specific effects on sensory stimuli. To do this they investigate the cellular, physiological and structural substrates that enable myoinhibitory peptide (MIP) to differentially modulate olfactory input to distinct olfactory glomeruli in the Drosophila brain.

Sizemore et al provide thorough and rigorous characterization of MIP-expressing neurons and their targets in the antennal lobe. While it doesn't directly explain how MIP is necessary and sufficient to simulate the drive of Drosophila toward food odors with the context of satiation, or how it influences odor preferences post-mating, it provides functional electrophysiology and imaging data describing how MIP influences olfactory glomeruli activity. Very few studies provide the functional data that explains how a peptide influences multiple olfactory glomeruli and this manuscript provides an important framework for understanding how neuropeptides can influence ethologically-relevant behaviors. This manuscript is an important addition to the neuromodulatory literature as it can be used as a model for not only how other peptides influence sensory behavior in Drosophila, but can also be extended to other, more complex brains since so little is understood about how peptides innervate and affect the function of discrete brain regions.

I have very few comments as I found the manuscript thorough, understandable and the language clear and concise. The methods are thorough and sufficient to replicate the work. The supplemental data is extensive and adds context and rigorous analysis to the main body figures and text. I also applaud the authors on their rigorous representation of raw data in all of the figures, use of color-blind friendly images, and for providing the crossing schemes for the more complicated genetics performed.

Major comments:

The only data the authors do not provide to support their hypothesis is a demonstration of the direct role of endogenous MIP in the MIP neurons. This would be possible by activating the MIP neurons (via olfactory stimulus or artificially if feasible) when MIP levels are reduced (for example by RNAi) then demonstrating this affects the functional role of MIP on post-synaptic neurons. However, the authors state that they tried to do a similar experiment, but were unable to detect any change in the postsynaptic neurons. Since the authors show rigorous and convincing data demonstrating how MIP influences activity of antennal glomeruli, and a functional role for the MIP receptor (sex-peptide receptor), I believe this renders demonstrating the endogenous MIP effects unnecessary and out of the scope of the current manuscript. However, should the authors want to determine the causal and/or coordinated effects of co-transmission of the MIP neurons on behavior or physiology in a future project/manuscript, this might be a helpful experiment.

> We thank the reviewer for this very helpful and constructive comment. The reviewer

raises an important point, which we have attempted to address in this revision. We agree with the reviewer that resolving the contribution of MIPergic LN-derived MIP, and if/how MIPergic LN-mediated co-transmission expands the role MIPergic LNs play in olfactory processing, is out of the scope of the current manuscript and is better suited for future research. We have added text to clarify interpretations based on features of the LNs relative to interpretations based on the exogenous application of MIP.

Minor comments:

1) *The use of ‘olfactory channels’ in the intro is confusing – easy to get it confused with olfactory receptors. Recommend rephrasing to be more specific (I think antennal glomeruli will work in this context).*

> Changed. This sentence (lines 43-44) now reads, “However, MIP appears to be uniformly distributed across all antennal lobe (AL) glomeruli, which process far more than just food-related odors.”

2) *Please clarify whether there were individual differences in SPR-Gal4 neuron expression similar to that seen with MIP neuron expression, or whether the expression of SPR-expressing neurons was consistent from fly to fly. I understand this might be difficult since the T2A line will show all neurons where SPR gene is expressed and SPR protein might be expressed, whereas your analysis with MIP antibody demonstrates where MIP protein is expressed. Clarifying this might be helpful to the reader.*

> The reviewer raises an important point, which we have clarified in this revision. To be clear, we never observed individual differences in SPR-T2A-GAL4 neuron expression based on our SPR-T2A-GAL4 MCFO analyses. To help clarify that point, we now include the following statement in this section of the Results (lines 305-307): “Notably, SPR-T2A-GAL4⁺ neurons were consistent across all animals – a stark contrast to the animal-to-animal differences in individual MIPergic LN morphology previously observed (**Fig. 2**).”

Additionally, we have included the modifier “consistent” in this part of the Results section where applicable. For example, in lines 296-298: “Within the brain, we find consistent colocalization of SPR-T2A-GAL4 and the proneural gene *embryonic lethal abnormal vision* (ELAV) (**Fig. 7h**) and the glial marker *reverse polarity* (REPO) (**Supplementary Fig. 4**).”

These changes are in addition to similar statements that were in our initial submission, such as in lines 371-375: “However, despite this variability, overall neuronal circuit functions persist including..., and SPR expression (**Fig. 7**).”

Reviewer #3 (Remarks to the Author):

In the manuscript by Sizemore and colleagues, the authors perform an impressive series of experiments to characterize the function of the MIP neuropeptide and its receptor SPR in the Drosophila antennal lobe (AL). MIP can modulate olfactory behaviors and the authors show it can modulate responses from olfactory neurons. Using co-staining with MIP antibodies and genetic labeling of AL-innervating neurons, the authors identify ~9 GABAergic LNs as the only neurons with the AL circuit to express MIP. By using mosaic labeling of these 9 MIP LNs, they identify them as patchy LNs. They further identify the innervation pattern of all MIP LNs and find they, like other patchy LNs, differ in their innervation patterns from animal to animal but as a whole still cover the entire AL. The authors present a comprehensive analysis of the glomerular input/output of the MIP neurons and do not identify differences that could explain the functional differences observed. They next use their anatomical identification of MIP LNs to query the EM reconstruction of the Drosophila brain, and identify 14 putative MIP neurons in this reconstruction. This affords them additional levels of analyses to even more comprehensively characterize the input/output relationships of MIP LNs. Their analyses again support the hypothesis that MIP neuron innervations cannot explain differential responses to MIP, prompting the authors to generate a new knockin of the SPR gene to capture its expression pattern. This reveals SPR heterogeneous expression in OSNs, LNs, and PNs, suggesting differential responses to MIP are due to differential expression of SPR. This is supported by RNAi knockdown of SPR in OSNs, which affected MIP-mediated functional effects observed in some but not all glomeruli.

This is a very impressive body of work aimed at investigating how an apparently ‘simple’ neuromodulatory system with a single neuropeptide and a single receptor affects the function of olfactory signaling. The data is robust and comprehensive. The methods are well explained and appropriate. The writing is clear and easy to follow. The genetic methods and steps employed to uncover the function of MIP are likely to become a guide for future investigators wishing to study the anatomy and function of other neuromodulators. I am enthusiastic about this study, and have only minor concerns that can be addressed without the need of additional experiments.

1. Figure 1a. Just visually comparing pre vs MIP injection GCAMP6 signals in DM1 and DM4, it appears odor responses go down not up as reported? Is the significant increase in response shown in (b) due to a decrease in basal fluorescence so that the ΔF increases? Or perhaps a more representative image should be shown?

> We thank the reviewer for this insightful point, and agree that the images formerly used in Figure 1a poorly represent the underlying data. We include images more representative of the underlying data in this revision.

Minor.

1. Please define MIP in the Figure 1 legend for its first appearance in figures (instead of in Figure 2 legend).

> Changed. The Figure 1 legend now reads: “**Fig. 1 | Myoinhibitory peptide (MIP) differentially modulates OSN ACV responses.**”

2. Page 2, right column. Please define the first use of MIP-ir here (and not on page 4, top left column as it is now).

> Changed. This sentence (lines 78-81) now reads, “For instance, there may be no synaptic input to DM1 OSNs from MIP-immunoreactive (MIP-ir) AL neurons, and therefore our prior observations (**Fig. 1b & 1c**) are the result of polysynaptic influences induced by MIP application.”

3. Page 6, bottom of left column/top of right column. “Thus, we first used several criteria (see Methods) to identify fully reconstructed putMIP LNs,” Given the importance of these criteria in the interpretation of the identified putMIPs, please summarize these criteria here in the main text. I.e, the criteria included identifying patchy LNs that receive synaptic inputs from CSB neurons, y, z, etc.

> Changed. We now include a brief summary of the criteria used to identify ideal putMIP LN candidates here in the main text (as well as in the figure legend and methods section). We now include the following statement on lines 188-192: “Thus, we first used several criteria to identify fully-reconstructed putMIP LNs, such as the candidates’ principal identity, previous AL LN subtyping results²⁸, synaptic connectivity, and neuromorphic similarity to R32F10-GAL4 AL LNs (**see Methods**) (**Fig. 3f**). These stringent criteria resulted in the identification of 14 ideal candidates (**Fig. 3f and Supplementary Table 1**).”

4. 9 MIP neurons were identified by genetic labeling, and 14 putative MIP neurons were identified in the EM reconstruction based on criteria mentioned above. Does this mean 5/14 are likely not MIP neurons? Or that additional MIP neurons might be present that were not genetically labelled? Some discussion on the difference in numbers would be helpful.

> The reviewer raises an important point, which we have hopefully clarified in this revision. For clarity, using antibody labeling with a primary against MIP we find that there are 8.7 ± 0.3 MIP-immunoreactive AL neurons (**Fig. 2d**). Therefore, the reviewer is correct that 5 of the 14 putative MIP neurons we have identified in the EM dataset are likely not MIP AL LNs. However, there are no methods currently available to objectively identify which 5 putative MIP LNs should be excluded. We therefore chose to analyze all candidates based on the stringent criteria used here, which are all supported by the known properties of MIPergic AL LNs. Even so, we generally find similar results for all 14 ideal candidates, such as the lack of synaptic polarity compartmentalization (**Fig. 3g**), each candidate generally avoids synaptically connecting to other candidates (**Fig. 4b and Supplementary Fig. 3j & 3k**), and every candidate receives far more excitatory vs. inhibitory input within a given AL glomerulus (**Fig. 4b**). Thus, while there are indeed 5 non-MIPergic AL LNs included within these analyses, we do not find significant distinguishing characteristics of any 5 of these putative MIP LNs that suggests our

reported interpretations are incorrectly attributable to their inclusion. Nevertheless, we have taken steps to explicitly acknowledge these “known-unknowns”, and only ever refer to these EM-reconstructed LNs as “putative MIPergic LNs (putMIP LNs)” accordingly.

To clarify this point, we now include the following statement on lines 600-612: “Despite these stringent criteria, these procedures resulted in the identification of 5 more neurons than there are MIP-immunoreactive neurons in the *Drosophila* AL (14 ideal candidates vs. ~9 MIP-immunoreactive neurons). This means that 5/14 putative MIP LNs analyzed here are likely not MIP AL LNs. As all known biological characteristics of MIPergic AL LNs were exhausted to identify these 14 ideal candidates, there is no convincingly reasonable method to identify which 5 putative MIP LNs should be excluded. Therefore, we elected to analyze all 14 ideal candidates (as opposed to imposing an arbitrary criteria) to ensure any/all MIPergic AL LN connectomic topological properties would be captured. That said, although ~36% of the putMIP LNs analyzed here are likely not MIPergic AL LNs, we generally find similar results for all 14 ideal candidates (**Figs. 3 & 4 and Supplementary Fig. 3**). Thus, we do not find significant distinguishing characteristics of any 5 of these putative MIP LNs that suggests our reported interpretations are incorrectly attributable to their inclusion.”

5. Figure 5. Please mention the genotype used for these experiments in the main text or figure legend.

> We have included the genotype of the animals used for this figure in the same manner as was done for figures 1 and 8.

6. Page 9, left column. Please define the first use of “DCV” here.

> Changed. This sentence now reads: “To determine which of these downstream partners (**Fig. 6b**) are likely targeted by MIPergic modulation, we determined which postsynaptic partners were downstream of putMIP LN terminals where dense core vesicles (DCVs) are observable (**Fig. 6c**).”

7. SPR is described as an ‘inhibitory receptor’. Please clarify somewhere in the main text (intro, discussion, etc.) what is known regarding how it functions as an inhibitory receptor. For example, is it a ligand-gate ion channel? Or does it trigger Gi signaling?

> We thank the reviewer for this clarification request. To be clear, SPR was previously misexpressed in Chinese hamster ovary (CHO) cells along with aequorin (a Ca^{2+} reporter) and either no additional G-protein (endogenous $G_{\alpha-q}$ only) or one of three chimeric G-proteins ($G_{\alpha-q_s}$, $G_{\alpha-q_i}$, or $G_{\alpha-q_o}$) (Yapici et al., 2008, *Nature*). This group then measured luminescence changes from these CHO cells in response to multiple concentrations of ligand, and found that the greatest luminescence change occurs when the CHO cells are co-transfected with either $G_{\alpha-q_i}$ or $G_{\alpha-q_o}$. This result suggests that peptide-mediated activation of SPR induces the $G_{\alpha-i}$ and/or $G_{\alpha-o}$ pathway(s) to modulate cyclic adenosine monophosphate (cAMP). This evidence was later independently supported by Oh & colleagues, who monitored cAMP levels within a putative SPR-

expressing neural ensemble via the cAMP sensor “Epac1-camps” (Oh et al., 2014, *PLOS Biology*). In this way, the latter investigators showed that cAMP levels decrease within their neurons-of-interest in response to various concentrations of MIP, until ~50uM where the effect begins to saturate.

We now include the following brief description of the known mechanism by which SPR acts as an inhibitory receptor on lines 275-277: “To determine which downstream partners are subject to MIPergic modulation, we identified the AL neurons that express MIP’s cognate receptor, the $G_{\alpha-i}/G_{\alpha-o}$ -coupled sex peptide receptor (SPR)^{79–82}.”

REVIEWER COMMENTS

Reviewer #1 (Remarks to the Author):

The authors have offered a revised version of their manuscript - though their adjustments fall far short of what I would've liked to see for this manuscript, I acknowledge they have addressed the issues I noted in the original review and can progress to publication.

I would, however, like to see the point regarding different focal planes and different genotypes accounting for the difference in the control values between Figures 1 and 8 referenced in the actual manuscript, not just the rebuttal response.

Reviewer #2 (Remarks to the Author):

In their revisions, Sizemore et al have done a commendable job of responding not only to my minor requests. Furthermore, after going through the other reviewer's requests, I feel they have sufficiently addressed comments from all reviewers. This paper is an important contribution to the literature, and although focused largely on anatomical characterization, the functional experiments provide important functional context to the role of MIP in olfactory response. All techniques have limitations and the authors appropriately use the techniques here to provide information about how a neuropeptide can potentially influence activity of neurons within the olfactory system. The thorough characterization sets up future directions very nicely as it can be later complemented by in vivo physiology in behaving animals.

Reviewer #3 (Remarks to the Author):

The revised manuscript includes updates to figures, new supplemental figures, updates to the main text, and an expanded discussion section that improves the quality and presentation of their findings. The revised manuscript sufficiently addresses my prior minor concerns.

This is an impressive body of work and a model of neural circuit analyses in the emerging field of fly connectomics. I have no further comments, and wish to congratulate the authors on their achievements.

Reviewer #1 (Remarks to the Author):

The authors have offered a revised version of their manuscript - though their adjustments fall far short of what I would've liked to see for this manuscript, I acknowledge they have addressed the issues I noted in the original review and can progress to publication.

I would, however, like to see the point regarding different focal planes and different genotypes accounting for the difference in the control values between Figures 1 and 8 referenced in the actual manuscript, not just the rebuttal response.

> We thank the reviewer for their agreeing that we have addressed the issues they noted in their original review and that we can progress to publication.

We now include the following statement on lines 854-862: "It is notable that subtle differences exist between the "pre-MIP"/baseline peb-GAL4 GCaMP responses (**Fig. 1**) and peb-GAL4>SPR-RNAi GCaMP responses (**Fig. 8**). The difference in baseline across these figures likely stems from: (1) subtle differences in the imaging plane which is inherent to these types of *in vivo* recordings across these experiments; and, (2) the difference in genotype of the animals being recorded in each respective figure. To address this, the within-figure baseline values are only ever internally compared by having within-animal measures across treatments (i.e., pre-MIP, MIP, and post-MIP). In doing so, the impact of MIP is with relation to the values from the same animal."

Reviewer #2 (Remarks to the Author):

*In their revisions, Sizemore et al have done a commendable job of responding not only to my minor requests. Furthermore, after going through the other reviewer's requests, I feel they have sufficiently addressed comments from all reviewers. This paper is an important contribution to the literature, and although focused largely on anatomical characterization, the functional experiments provide important functional context to the role of MIP in olfactory response. All techniques have limitations and the authors appropriately use the techniques here to provide information about how a neuropeptide can potentially influence activity of neurons within the olfactory system. The thorough characterization sets up future directions very nicely as it can be later complemented by *in vivo* physiology in behaving animals.*

> We thank the reviewer for their compliments, and for the service & attention they have afforded us while serving as a reviewer.

Reviewer #3 (Remarks to the Author):

The revised manuscript includes updates to figures, new supplemental figures, updates to the main text, and an expanded discussion section that improves the quality and

presentation of their findings. The revised manuscript sufficiently addresses my prior minor concerns.

This is an impressive body of work and a model of neural circuit analyses in the emerging field of fly connectomics. I have no further comments, and wish to congratulate the authors on their achievements.

> We thank the reviewer for their compliments, and for the service & attention they have afforded us while serving as a reviewer.

REVIEWERS' COMMENTS

Reviewer #1 (Remarks to the Author):

The authors have addressed my additional concern. Thank you.

Reviewer #1 (Remarks to the Author):

The authors have addressed my additional concern. Thank you.

> We are pleased that we could address the reviewer's additional concern, and would like to thank them for the service & attention they have afforded us while serving as a reviewer.